# Retinal Processing: Insights from Mathematical Modelling

**DOI:** 10.3390/jimaging8010014

**Published:** 2022-01-17

**Authors:** Bruno Cessac

**Affiliations:** France INRIA Biovision Team and Neuromod Institute, Université Côte d’Azur, 2004 Route des Lucioles, BP 93, 06902 Valbonne, France; bruno.cessac@inria.fr

**Keywords:** retinal network, visual system, spatio-temporal spike correlations, linear response, non stationarity

## Abstract

The retina is the entrance of the visual system. Although based on common biophysical principles, the dynamics of retinal neurons are quite different from their cortical counterparts, raising interesting problems for modellers. In this paper, I address some mathematically stated questions in this spirit, discussing, in particular: (1) How could lateral amacrine cell connectivity shape the spatio-temporal spike response of retinal ganglion cells? (2) How could spatio-temporal stimuli correlations and retinal network dynamics shape the spike train correlations at the output of the retina? These questions are addressed, first, introducing a mathematically tractable model of the layered retina, integrating amacrine cells’ lateral connectivity and piecewise linear rectification, allowing for computing the retinal ganglion cells receptive field together with the voltage and spike correlations of retinal ganglion cells resulting from the amacrine cells networks. Then, I review some recent results showing how the concept of spatio-temporal Gibbs distributions and linear response theory can be used to characterize the collective spike response to a spatio-temporal stimulus of a set of retinal ganglion cells, coupled via effective interactions corresponding to the amacrine cells network. On these bases, I briefly discuss several potential consequences of these results at the cortical level.

## 1. Introduction

Let us start with a very simple experiment. Look around you... That’s it, the experiment is over. A very ordinary experience, isn’t it? Is it really though? Let us first point out that looking around you to see, that is, having the sense of sight, is indeed ordinary—except for those who have partially or totally lost their ability to see. We will come back to this point at the end of the paper. Now, excluding visual impairments, vision is everything but ordinary.

Think of it. A flux of photons, with frequencies in the visible spectrum range, emitted by the external world around us enters into our eyes, then “something” happens, and we see. Thanks to constant progress in experimental and theoretical neuroscience, we understand better and better this “something”, the mechanisms of vision, although our view of it is far from being complete. In particular, in these times of artificial intelligence, bio-inspired computing, computer vision, it might be helpful to understand how our brain is able to handle the complex visual information coming from the external world so rapidly and efficiently with an energy consumption of the order of a few Watts.

Certainly, the retina plays a central role in this process. It has been known for long time that this is definitely not a camera. The retina is smart [1] and it has to be. Think especially of the difference of scale between the retina and the visual cortex, in terms of size but also numbers of neurons and synapses. As everything that goes to the visual cortex comes from the retina, this little membrane, at the back of the eye, half a millimetre thick, with an area of order a cm${}^{2}$ (for humans), has to some extent to filter the visual information, leaving out “irrelevant details” and capturing crucial events, and then, signal them appropriately to the brain via spike trains. As a matter of fact, the question(s) of “efficiently” encoding information by spikes has been the subject of many fascinating papers [2,3,4], especially in the seminal paper from Barlow [5] with concepts such as reducing redundancy, information compression and efficient coding. These concepts are regularly updated with recent experimental and theoretical investigations [6,7,8,9,10,11,12,13,14,15]. We come back to this point at the end of the paper too.

The retina has, roughly, the following structure. For more details, see, e.g., [16] or https://webvision.med.utah.edu/book/part-i-foundations/simple-anatomy-of-the-retina/ (accessed on 22 November 2021). It is organized in five neuronal types: photo-receptors, rods and cones (P), horizontal cells (H cells), bipolar cells (B cells), amacrine cells (A cells), retinal ganglion cells (RG cells), to which are added glial cells (Mueller’s cells). These neuron types are connected by chemical and electric synapses, in specific functional circuits or “pathways” (like the rod–cone pathway [17,18]), which are a key in the retinal capacity to convert the light coming from a visual scene into spike patterns sent to the visual cortex, through the Lateral Geniculate Nucleus (LGN), via the optic nerve made of RG cells axons. In particular, in the retina, there are very specific synapses like the ribbon synapse enabling neurons to transmit light signals from photoreceptors to B cells over a dynamic range of several orders of magnitude in intensity [19]. Roughly, two main connectivity structures can be distinguished: feed-forward, the P-B-G path which leads from the photo transduction to the spike trains emitted by the RG cells towards the cortex. There is also a lateral connectivity through H cells, at the origin of the Center-Surround structure of the receptive fields, and the A cells whose role is still poorly understood and which are one of the main objects of study of this paper.

The structure of the retina and its behaviour are thus well studied on the experimental side. There are comparatively fewer modelling studies although important work has been done on retinal coding [20,21,22,23,24], biophysically detailed models [25,26,27] and generalized linear models applied to retinal coding [28,29,30]. Several powerful software has been designed to model the retina at different scales such as COREM [31], Convis [32], Isetbio https://github.com/isetbio/isetbio/wiki (accessed on 22 November 2021). The Virtual Retina simulator, developed by A. Wohrer and P. Kornprobst [33] at INRIA, was one of the first of these simulators and has given rise to subsequent simulators in our group, the platform PRANAS [34], https://team.inria.fr/biovision/pranas-software/ (accessed on 22 November 2021) and more recently Macular https://team.inria.fr/biovision/macular-software/ (accessed on 22 November 2021). There are quite less mathematical results on how retinal structure, especially lateral A cells’ connectivity, shapes the spike response to spatio-temporal stimuli [35,36,37].

One of the goals of this paper is to elicit reflections in this direction, grounded on mathematical developments fed by the recent progress in the knowledge of retina physiology and structure. This is a humble and partial point of view, resulting from my collaboration with neurobiologists experts in the retina. The paper contains new results, essentially the mathematically tractable model of the layered retina integrating amacrine cells lateral connectivity and the mathematical framework to handle piecewise linear rectification presented in Section 2, the study of rectification effects on retinal ganglion cells receptive field (Section 3.1), the study of voltage and spike correlations of retinal ganglion cells (Section 3.2) and the discussion about the mixed effect of network and stimulus on spike correlations in Section 3.3. It also contains already published material, essentially the framework and results dealing with Gibbs distributions and linear response (Section 2.2.1 and Section 3.3).

The goal is to draw a common thread about the potential role of amacrine cells from retinal spatio-temporal stimuli response to spike coding. More precisely, I am addressing the following problems on mathematical grounds. In the main text, I focus on the neuroscience modelling perspective, whereas, in the Conclusions section, I discuss potential consequences of these results out of the field of neuroscience.

***Problem 1.*** *How does the structure of the retina, in particular, amacrine lateral connectivity, condition the retinal response to dynamic stimuli?*

The problem can be addressed at two levels. *Level 1. Single cell response to stimuli.* The individual response of ganglion cells is usually expressed in terms of their receptive field. This notion is on the one hand phenomenological: it is observed that each ganglion cell responds preferentially to stimuli, localized in space, with a characteristic spatio-temporal structure. For example, an ON-Center cell preferentially responds to an increase in luminance in a circular area corresponding to the central part of the receiving field. This notion is also expressed mathematically by a kernel $K_{G_{}}$, i.e., a function of space and time, so that the response of an RG cell to a spatio-temporal stimulus S(x,y,t), takes the form:(1)$$\left[\;K_{G_{}}\overset{x,y,t}{*}S\;\right]\left(t\right)=\int_{x=-\infty }^{+\infty }\;\int_{y=-\infty }^{+\infty }\;\int_{s=-\infty }^{t}\;K_{G_{}}\left(x-x_{C},y-y_{C},t-s\right)\;S\left(x,y,s\right)dx\;dy\;ds,$$
where $\overset{x,y,t}{*}$ means space (x,y)-time (*t*) convolution. $x_{C},y_{C}$ are the coordinates of the RF center. The integrals are well defined since the kernel decreases fast enough to infinity, in space and time, to guarantee convergence. The upper bound in time, *t*, expresses causality, whereas the lower bound, −∞, implicitly assumes that the stimulus has been applied in a distant past compared to *t*, quite longer than the characteristic times involved in RG cell response.

Equation (Equation 1) corresponds to a linear response. It is therefore only valid for stimuli of low amplitude in voltage. More generally, the voltage response to the stimulus is a functional of the stimulus that one can, under well posed mathematical conditions, write as a Volterra expansion [21], (Equation 1) being the lowest order (linear) term. Unfortunately, higher-order terms are essentially inaccessible experimentally and one usually constrains instead the nonlinearity of the response under other modalities. In particular, taking into account that the response of a ganglion cell to a stimulus is, ultimately, a sequence of spikes, one writes the probability density of emitting a spike between *t* and t+dt in the form $f\left(\;\left[\;K_{G_{}}\overset{x,y,t}{*}S\;\right]\left(t\right)+b\;\right)$, where *f* is a nonlinear positive increasing function (typically, a sigmoid), and *b* is a threshold constraining the level of activity of the RG cell in the absence of stimulation. This procedure defines an inhomogeneous Poisson process called the linear-nonlinear Poisson (LNP) model [38,39,40]. Experimentally, the kernel $K_{G_{}}$ is determined by Spike-Triggered Average or Spike-Triggered Correlation technique, studying the response to a white noise [38]. Nonlinearity is then determined, typically by the Levenberg–Marquardt method [41]. This modelling asks, however, the following questions:(i)How is the kernel $K_{G_{}}$ of the RG cells constrained by the structure/dynamics of the upper layers of retinal cells?(ii)The forms (Equation 1) implicitly assumes that $K_{G_{}}$ does not depend on the stimulus. Can one write mathematical conditions that guarantee such an independence?(iii)To which extent is the notion of Ganglion cells Receptive Field compatible with nonlinear effects reported in retinal neurons and synapses, such as voltage rectification or gain control?

*Level 2. Collective response to stimuli and spike statistics.* RG cells do not interact directly, but amacrine connectivity induces an effective interaction between them. What is therefore the structure of the spatio-temporal correlations induced by the conjunction of the spatio-temporal stimulus and the response of the retinal network, in particular, the amacrine lateral connectivity? A classical paradigm in neural coding is to assume that the retina decorrelates RG cell outputs to maximize information transfer [6,7,8,9,10,11,13,14,15]. It is in particular believed that A cells play a central role in this decorrelation process (see [15] and references therein). What can be, at the mathematical level, the conditions, on the stimulus and dynamics that allow a network of neurons interacting with each other to produce vanishing, or at least, *weak* correlations? When does *weak* mean *negligible*? These questions are actually closely related to the second problem.

***Problem 2.*** *How do retina network and dynamics shape spike statistics in the response to stimuli?*

More generally, considering the retina as a dynamical system forced by non-stationary, spatially inhomogeneous stimuli, what could be a general form for the (non-stationary) statistics of spike trains emitted by ganglion cells, taking into account that spike trains emitted by the retina are all that the LGN and cortex see? One can attempt to construct a canonical form of probability distributions of the retinal spike trains taking into account that:(i)Stimuli, thus statistics, are not stationary;(ii)The cortex (and, before, the LGN) only receive spikes, thus have no information about the biophysical processes which have generated those spikes and no information on the underlying dynamics of the retina (voltages, activation variables, conductances). All the information is contained in the spatio-temporal structure of spikes;(iii)Spike train distributions may exhibit long time scale dependence (i.e., have a long memory).

In this paper, I address these problems with the help of two models. The first, presented in Section 2.1, grounded on biology and e.g., the papers [42,43,44,45] mimics the Bipolar-Amacrine-Ganglion cells network and is used, in Section 3.1 and Section 3.2, to make progress in elucidating problem 1. I first show how one can obtain an explicit form for the kernel (Equation 1) featuring the A cells lateral connectivity. This RF explicitly depends on the BC cells-A cells network through the eigenvalues and eigenvectors of an operator I call “transport operator”. I discuss some consequences of this result, especially in terms of response to propagating stimulus. This result is valid when cells act as linear integrators. However, cells are in general rectified by nonlinearities. I propose piecewise-linear rectifications (as used in several retina model) and I discuss how rectification acts on the RF of Equation (Equation 1). A striking conclusion is that, if the convolution from (Equation 1) is preserved, this is the price of having a RF depending on the stimulus. A consequence of this analysis is that spike correlations may depend on the stimulus and are expected to be quite different when considering e.g., objects moving along trajectories in comparison to static images.

The second model, introduced in Section 2.2 and analysed in Section 3.3, attempts to propose a canonical form of probability distributions of the retinal spike trains based on the constraints (i), (ii), (iii) above. These sections essentially present the conclusions of works published elsewhere [46,47,48,49,50,51]. As I argue, these constraints lead to a natural notion of spike probabilities, somewhat extending the statistical physics notion of Gibbs distribution to the non-stationary case. In this setting, one establishes a linear response for a network of interacting spiking cells that can mimic a set of RG cells coupled via effective interactions corresponding to the A cells network influence. This linear response theory not only gives the effect of a non-stationary stimulus to first order spike statistics (firing rates) but also its effect on higher order correlations. Indeed, spike correlations are modified by a spatio-temporal stimulus and can be computed thanks to the knowledge of spontaneous correlations. The linear response formula is expressed as a convolution where the kernel can be explicitly computed for an Integrate and Fire conductance based model [51]. Moreover, as I argue, these spike train distributions have close links with information geometry. In particular, they induce a natural metric in an abstract space of probabilities, with close potential links with the neuro-geometry introduced by Sarti, Citti, and Petitot et al. [52,53,54,55]. This is discussed in the Conclusions section.

More generally, the application and discussion sections shortly propose the possible extension of this work to several domains: retinal prostheses, Section 4.1; Convolutional networks, Section 4.2; Implications for cortical models, Section 5.1; and Neuro-geometry, Section 5.2.

## 2. Materials and Methods

### 2.1. Modelling the Retinal Network

#### 2.1.1. Specifics of the Retina

Neurons in the retina have the same biophysics as their cortical counterparts. However, they operate under different modalities. Remarkably, with the exception of the RG cells, the retinal neurons do not emit action potentials. Their activity and interactions therefore take place through graded (continuous) membrane potentials as opposed to the sharp peak of an action potential. Furthermore, there is no long-term synaptic plasticity in the retina. Finally, the main “computational” elements in the retina are functional circuits [18] made of a few neurons and synapses, in large contrast with “computational” units in the visual cortex, such as cortical columns, involving thousands of neurons. A modelling consequence is that mean-field or neural masses description used in the cortex might not be relevant to study the retina.

The goal of this paper is to address mathematical questions about the dynamics and behaviour of the retina embedded in the visual system. To instantiate these questions on a firm mathematical ground, we are going to consider a model of the retinal network, based on a few fundamental facts briefly exposed in the previous section:The retina is a high dimensional, non autonomous and noisy dynamical system, layered and structured, with non-stationary and spatially inhomogeneous entries (visual scenes).Most retinal neurons are not spiking, except RG cells. Thus, the retina performs analogic computing.Local retinal circuits efficiently process the local visual information. These local circuits are connected together, spanning the whole retina in a regular tiling. From this perspective, it is important to consider individual neurons and synapses, in contrast, e.g., to cortical modelling, where it is relevant to consider mean-field approaches averaging over populations.

Thus, the model presented below and in Figure 1 is non-stationary, with a layered retina like structure, where dynamics ruling B cells, A cells, and RG cells voltage are piecewise linear. As we discuss, the model affords additional nonlinearities like gain control. For RG cells, the spiking process is mimicked by a nonlinear firing rate so that our model enters in the class of LNP models.

#### 2.1.2. Structure of the Retina Model

We assimilate the retina to a superimposition of 3 layers, each one being a flat, two-dimensional square of edge length *L* mm where spatial coordinates are noted x,y (Figure 1). Each layer corresponds to a cell population (B cells, A cells, RG cells) where the density of cells is taken to be uniform. We note $\delta_{p}$ the lattice spacing in mm, and $N_{p}$ the total number of cells in the layer *p*. Without loss of generality, we assume that *L*, the retina’s edge size, is a multiple of $\delta_{p}$. We note $L_{p}=\frac{L}{\delta_{p}}$, the number of cells *p* per row or column so that $N_{p}=L_{p}^{2}$. Each cell in the population *p* thus has Cartesian coordinates $\left(x,y\right)=\left(i_{x}\delta_{p},i_{y}\delta_{p}\right)$, $\left(i_{x},i_{y}\right)\in {\left\{\;1,\cdots ,L_{p}\;\right\}}^{2}$. To avoid multiples indices, we associate to each pair $\left(i_{x},i_{y}\right)$ a unique index $i=i_{x}+\left(i_{y}-1\right)\;L_{p}$. The cell of population *p*, located at coordinates $\left(i_{x}\delta_{p},i_{y}\delta_{p}\right)$, is then denoted by $p_{i}$.

One can roughly subdivide the real retina into two blocks (Figure 1). The first we name in short, for modelling purposes OPL, (Outer Plexiform Layer), includes the P, H cells, B cells, and the related synapses. (Note that the terminology OPL and IPL actually refers to synaptic layers. “The outer plexiform layer has a wide external band composed of inner fibres of rods and cones and a narrower inner band consisting of synapses between photoreceptor cells and cells from the inner nuclear layer.” “The inner plexiform layer consists of synaptic connections between the axons of bipolar cells and dendrites of ganglion cells” (ref https://www.sciencedirect.com/topics/medicine-and-dentistry/, accessed on 22 November 2021). In our model, these naming are short cuts to distinguish the network input (OPL) and the network processing (IPL)). An “input” of this block is the flow of photons emitted by the outside world and picked up by the photo-receptors. In our model, this corresponds to a “stimulus”, i.e., a function S(x,y,t) where x,y are (two-dimensional) space coordinates and *t* is the time coordinate. As we do not consider color sensitivity here, S characterizes a black and white scene, with a control on the level of contrast ∈[0,1]. The “output” of the OPL is sent to B cells in the form of a “drive” voltage, defined in Equation (Equation 2) below. In the real retina, the voltage of each BCell integrates, spatially and temporally, the local visual information of the photo-receptors which are connected to it, with a lateral modulation due to the H cells. Each B cells is thus sensitive to specific local characteristics of the visual scene, defining its Receptive Field (RF). Thus, B cells, like RG cells, have a receptive field. However, as they are earlier in the vertical pathway, they integrate less features. Note that the RF of distinct B cells usually overlap creating correlations between B cells voltages (see Section 3.2.1).

We label B cells (layer 1) with the index $i=1,\cdots ,N_{B}$ and we model the RF of B cells by a convolution kernel, $K_{B_{i}}$, such that the voltage of BCell *i* is stimulus-driven by the term:(2)$$V_{i_{drive}}\left(t\right)=\left[\;K_{B_{i}}\overset{x,y,t}{*}S\;\right]\left(t\right).$$

The center of the RF, located at $x_{i},y_{i}$, also corresponds to the coordinates of the BCell *i*. A typical shape for the RF of B cells is illustrated in Figure 2, although the explicit form does not play a role in the subsequent mathematical developments.

The second block that we name in short IPL (Inner Plexiform Layer) comprises the A cells and RG cells and the afferent synapses. Its “input” is the output of the OPL, and its output the trains of action potentials emitted by the RG cells. A cells are difficult to study experimentally because they are hardly accessible from electrophysiology measurements. There are also a large number of cell subtypes in the A cells class (around 40), of which only a small number have duly identified functions. It is, however, recognized that they play an essential role in the treatment of motion [17,56,57]. Here, we address mathematically the question of the RG cell receptive field form, resulting from the pooling of B cells, as illustrated in Figure 1, each with a specific RF as exemplified in Figure 2, and modulated by A cells lateral connectivity.

#### 2.1.3. B Cells–A Cells Interactions

We label A cells (second layer) with the index $j=1\cdots N_{A}$. We note $W_{B_{i}}^{A_{j}}$ as the synaptic weight from A cell *j* to B cell *i* and $W_{A_{j}}^{B_{i}}$ the synaptic weight from B cell *i* to A cell *j*. We set $W_{B_{i}}^{A_{j}}\leq 0$, (A cells are in general inhibitory although some excitatory A cells exist, not considered here), whereas $W_{A_{j}}^{B_{i}}\geq 0$. The synaptic weight matrices B cells to A cells and A cells to B cells are noted $W_{A_{}}^{B_{}},W_{B_{}}^{A_{}}$. They are not squared in general. Electric synapses (gap junctions) between B cells and A cells also exist (e.g., in the Rod Cone pathway [17], see also [58] and https://www.ncbi.nlm.nih.gov/books/NBK549947/ (accessed on 22 November 2021)), but we will not consider them in first place, for simplicity. Note, however, as briefly discussed in Section 2.1.8, that adding gap junctions would simply result in adding linear terms to Equations (Equation 3), (Equation 6) and (Equation 9) (when considering passive gap junctions) and modify characteristic time scales, without changing the global analysis.

The voltage of B cell *i*, $V_{B_{i}}$, evolves according to: (3)$$\frac{dV_{B_{i}}}{dt}=-\frac{1}{B_{i}}V_{B_{i}}+\sum_{j=1}^{N_{A}}W_{B_{i}}^{A_{j}}\;N_{A}\left(\;V_{A_{j}}\;\right)+F_{B_{i}}\left(t\right).$$

Here, $B_{i}$ is the characteristic time scale of B cell *i* response (in ms). The function:(4)$$N_{A}\left(V\right)=\left\{\begin{array}{cc} V-\theta_{A},\; & if\;\;V_{A}>\theta_{A};\\ 0,\; & \mathrm{otherwise} \end{array},\right.$$
is a linear rectifier ensuring that the synapse j→i becomes silent when the voltage of the pre-synaptic A cell *j*, $V_{A_{j}}$, is lower than a threshold $\theta_{A}$. This corresponds to a biophysical fact: a synapse cannot change its sign. For simplicity, we consider $\theta_{A}$ to be the same for all A cells, although the present formalism can be extended, e.g., to several families of A cells having different thresholds. Note that linear rectifiers of type (Equation 4) rectify cell’s voltage “from below”. Rectification “from above” also exist, ensuring that the cell’s voltage does not increase without bounds. A typical mechanism is gain control, where an additional variable, called the activity, increasing as voltage increases, triggers a gain function nonlinearly dropping down the voltage when it exceeds an upper threshold [42,44]. Under some mild assumptions, gain control can also be implemented as a linear function of the activity. This is discussed in Section 2.1.8 as an extension to the present model.

Finally, $F_{B_{i}}\left(t\right)$ is the OPL input term. To match classical retina models as developed e.g., in [42,44], it reads:(5)$$F_{B_{i}}\left(t\right)=\frac{V_{i_{drive}}}{\tau_{B}}\;+\;\frac{dV_{i_{drive}}}{dt}=\left[\;K_{B_{i}}\overset{x,y,t}{*}\left(\;\frac{S}{\tau_{B}}+\frac{dS}{dt}\;\right)\;\right]\left(t\right),$$
(where $K_{B_{i}}\left(x,y,0\right)=0$). In short, $F_{B_{i}}\left(t\right)$ is chosen so that, in the absence of A cells interaction, $V_{B_{i}}\left(t\right)=V_{i_{drive}}\left(t\right)$. Note that $F_{B_{i}}\left(t\right)$ implements therefore a time derivative of the drive, which makes, e.g., a B cell response to moving objects sensitive to changes in directions or speed.

A cells are connected to B cells with chemical synapses. The differential equation obeyed by the voltage of A cell *j* is: (6)$$\frac{dV_{A_{j}}}{dt}=-\frac{1}{A_{j}}V_{A_{j}}+\sum_{i=1}^{N_{B}}W_{A_{j}}^{B_{i}}\;N_{B}\left(\;V_{B_{i}}\;\right),$$
where $A_{j}$ is the characteristic time scale of A cell *j* response, and $N_{B}$ has the same form as (Equation 4), with a threshold $\theta_{B}$. Note that, in contrast to B cells, A cells do not receive an OPL input.

#### 2.1.4. RG Cells

We label RG cells (third layer) with the index $k=1\cdots N_{G}$. They are connected to B cells with excitatory synaptic weights, $W_{G_{k}}^{B_{i}}\geq 0$ (e.g., glutamatergic synapses) and to A cells with inhibitory synaptic weights, $W_{G_{k}}^{A_{j}}\leq 0$ (e.g., glycinergic or GABA-ergic synapses). Their voltage, $V_{G_{k}}$, evolves according to:(7)$$\frac{dV_{G_{k}}}{dt}=-\frac{1}{\tau_{G}}V_{G_{k}}+\sum_{i=1}^{N_{B}}W_{G_{k}}^{B_{i}}N_{B}\left(V_{B_{i}}\right)+\sum_{j=1}^{N_{A}}W_{G_{k}}^{A_{j}}N_{A}\left(V_{A_{j}}\right).$$

RG cells are spiking. In the model, their spiking activity (firing rate) is defined by an LNP model [38,39,40]. It depends on the voltage via a nonlinear function $N_{G}\left(\;V_{G}\;\right)\equiv f\left(\;\frac{V_{G}\left(t\right)-\theta_{G}}{\sigma_{G}}\;\right)$, where *f* is typically a sigmoid. Although the detailed form of *f* does not matter here, it will be convenient, in the sequel, to consider:(8)$$N_{G}\left(\;V_{G}\;\right)=\frac{1}{\sqrt{2\pi }}\int_{-\infty }^{\frac{V_{G}-\theta_{G}}{\sigma_{G}}}e^{-\frac{x^{2}}{2}}\;dx.$$

The parameters $\theta_{G}$ (spiking threshold) and $\sigma_{G}$ (controlling the slope of the sigmoid at $V_{G}=\theta_{G}$) corresponds, in the case where $N_{G}$ has the form (Equation 8), to the probability that a Gaussian centred Ornstein–Uhlenbeck processes with mean-square deviation $\sigma_{G}$ crossing the threshold $\theta_{G}$.

#### 2.1.5. Joint Dynamics

The joint dynamics of all cells voltage are given by the dynamical system:(9)$$\left\{\begin{array}{cccc} \frac{dV_{B_{i}}}{dt} & = & -\frac{1}{\tau_{B}}V_{B_{i}}+\sum_{j=1}^{N_{A}}W_{B_{i}}^{A_{j}}\;N_{A}\left(\;V_{A_{j}}\;\right)+F_{B_{i}}\left(t\right), & i=1\cdots N_{B};\\ \; & \; & \\ \frac{dV_{A_{j}}}{dt} & = & -\frac{1}{\tau_{A}}V_{A_{j}}+\sum_{i=1}^{N_{B}}W_{A_{j}}^{B_{i}}\;N_{B}\left(\;V_{B_{i}}\;\right), & j=1\cdots N_{A};\\ \; & \; & \\ \frac{dV_{G_{k}}}{dt}\; & = & -\frac{1}{\tau_{G}}V_{G_{k}}+\sum_{i=1}^{N_{B}}W_{G_{k}}^{B_{i}}N_{B}\left(V_{B_{i}}\right)+\sum_{j=1}^{N_{A}}W_{G_{k}}^{A_{j}}N_{A}\left(V_{A_{j}}\right), & k=1\cdots N_{G}; \end{array}\right.$$
whereas RG cells spikes are produced by the LNP mechanism described above.

The system of Equation (Equation 9) can be summarized as follows (Figure 1). B cells receive the visual input via the term $F_{B_{i}}\left(t\right)$ which depends on the stimulus and on the B cell’s receptive field. They are inhibited by A cells via the synaptic weights $W_{B_{i}}^{A_{j}}<0$. A cells are excited by B cells via the synaptic weights $W_{A_{j}}^{B_{i}}>0$. B cells are connected to RG cells via the synaptic weights $W_{G_{k}}^{B_{i}}>0$. A cells are connected to RG cells via the synaptic weights $W_{G_{k}}^{A_{j}}<0$. Note that we do not impose any constraint on the connectivity here.

To study mathematically the dynamical system (Equation 9), we write it in a more convenient form. We use Greek indices $\alpha ,\beta ,\gamma =1\cdots N\equiv N_{A}+N_{B}+N_{G}$, and define the state vector $\vec{X}$, with entries:(10)$$X_{\alpha }=\left\{\begin{array}{cccc} \; & V_{B_{i}},\; & \alpha =i, & i=1\cdots N_{B};\\ \; & V_{A_{j}},\; & \alpha =N_{B}+j, & j=1\cdots N_{A};\\ \; & V_{G_{k}},\; & \alpha =N_{B}+N_{A}+k, & k=1\cdots N_{G}. \end{array}\right.$$

We introduce $\vec{F}\left(t\right)$, the non-stationary input, with entries:$$F_{\alpha }\left(t\right)=\left\{\begin{array}{cccc} & F_{B_{i}}\left(t\right),\; & \alpha =i, & i=1\cdots N_{B};\\ & 0,\; & \alpha >N_{B}; \end{array}\right.$$
and $\vec{R}\left(\vec{X}\right)$, the rectification term, with entries:$$R_{\alpha }\left(\vec{X}\right)=\left\{\begin{array}{cccc} & N_{B}\left(\;V_{B_{i}}\;\right),\; & \alpha =i, & i=1\cdots N_{B};\\ & N_{A}\left(\;V_{A_{j}}\;\right),\; & \alpha =N_{B}+j, & j=1\cdots N_{A};\\ & 0,\; & \alpha =N_{B}+N_{A}+k, & k=1\cdots N_{G}. \end{array}\right.$$

We use the notation $0_{n_{1}n_{2}}$ for the $n_{1}\times n_{2}$ matrix with zero entries. We introduce the N×N matrices:(11)$$T=\left(\;\begin{array}{cccccc} \; & -\mathrm{diag}{\left[\;\tau_{B_{i}}\;\right]}_{i=1\cdots N_{B}} & \; & 0_{N_{B}N_{A}} & \; & 0_{N_{B}N_{G}}\\ \; & 0_{N_{A}N_{B}} & \; & -\mathrm{diag}{\left[\;\tau_{A_{j}}\;\right]}_{j=1\cdots N_{A}} & \; & 0_{N_{A}N_{G}}\\ \; & 0_{N_{G}N_{B}} & \; & 0_{N_{G}N_{A}} & \; & -\mathrm{diag}{\left[\;\tau_{G_{k}}\;\right]}_{k=1\cdots N_{G}} \end{array}\;\right),$$
characterizing the characteristic integration times of cells,
(12)$$W=\left(\;\begin{array}{cccccc} \; & 0_{N_{B}N_{B}} & \; & W_{B_{}}^{A_{}} & \; & 0_{N_{B}N_{G}}\\ \; & W_{A_{}}^{B_{}} & \; & 0_{N_{A}N_{A}} & \; & 0_{N_{A}N_{G}}\\ \; & W_{G_{}}^{B_{}} & \; & W_{G_{}}^{A_{}} & \; & 0_{N_{G}N_{G}} \end{array}\;\right),$$
summarizing chemical synapses interactions. Note that, to our best knowledge, there are no synapses from RG cells to RG cells, but they could be added in this formalism.

Then, the dynamical system (Equation 9) reads, in vector form:(13)$$\frac{d\vec{X}}{dt}=T^{-1}.\vec{X}+W.\vec{R}\left(\vec{X}\right)+\vec{F}\left(t\right).$$

We remark that (Equation 13) has a specific skew-product structure: the dynamics of RG cells is driven by B cells and A cells with no feedback. This means that one can study first the coupled dynamics of B cells and A cells and then the effect on RG cells. This corresponds to a biological reality as, to our best knowledge, there is no feedback from RG cells to B cells or to A cells.

#### 2.1.6. Piecewise Linear Evolution

We assume here that $F_{\alpha }\left(t\right)$ is bounded, as well as synaptic weights. Thus, the phase space Ω of (Equation 13) can be taken to be compact. Indeed, trajectories cannot escape to infinity thanks to the rectification terms $N_{B},N_{A}$, (Equation (Equation 4)) and thanks to the sign of synaptic weights $W_{A_{j}}^{B_{i}},W_{B_{i}}^{A_{j}}$. More precisely, $V_{B_{i}}$ cannot become arbitrarily large and positive because the input term $F_{\alpha }\left(t\right)\equiv F_{B_{i}}\left(t\right)$ is bounded and because $\sum_{j=1}^{N_{A}}W_{B_{i}}^{A_{j}}\;N_{A}\left(\;V_{A_{j}}\;\right)\leq 0$. Assume indeed that $V_{B_{i}}$ increases (due to a large enough $F_{\alpha }>0$ making the r.h.s. of Equation (Equation 3) positive). This leads to an increase of connected A cell voltages $V_{A_{j}}$ (Equation (Equation 6)), thus to a decrease of the term $\sum_{j=1}^{N_{A}}W_{B_{i}}^{A_{j}}\;N_{A}\left(\;V_{A_{j}}\;\right)\leq 0$ until the point where the r.h.s. of (Equation 3) becomes negative, thereby decreasing $V_{B_{i}}$ and preventing it from becoming arbitrarily large. This implies as well that $V_{A_{j}}$s cannot become arbitrarily large. On the opposite, if $V_{B_{i}}$ (resp. $V_{A_{j}}$) becomes smaller than $\theta_{B}$ (resp. $\theta_{A}$), it does not play any more role in the dynamics because of rectification.

Due to the specific form (Equation 4) of the rectification terms, the dynamical system (Equation 13) is piecewise linear. More precisely, we can partition the phase space Ω into sub domains $\Omega^{\left(n\right)}$, $n=1\cdots 2^{N_{B}+N_{A}}$ defined as follows. To each cell $\alpha =1\cdots N_{B}+N_{A}$ (B cell or A cell), we associate a “rectification label” $\eta_{\alpha }=1$ if the cell α is rectified and $\eta_{\alpha }=0$ otherwise. Because of the form (Equation 4) of the rectification, the label $\eta_{\alpha }$ corresponds to a partition of the voltage $X_{\alpha }$’s domain of variation into two sub domains (e.g., for a B cell, $\eta_{\alpha }=1$ if $V_{B_{i}}<\theta_{B}$ and $\eta_{\alpha }=0$ if $V_{B_{i}}\geq \theta_{B}$). Now, the set ${\left\{\;0,1\;\right\}}^{N_{B}+N_{A}}$ is made of chains $\eta =\left(\;\eta_{1}\cdots \eta_{N_{B}+N_{A}}\;\right)$ composed of the rectification labels $\eta_{\alpha }$ of all B cells and A cells. To each such sequence is therefore associated a convex domain $\Gamma^{\left(n\right)}$ of $R^{N_{B}+N_{A}}$ where all cells α such that $\eta_{\alpha }=0$ have their voltage $X_{\alpha }$ larger than the rectification threshold, thus, are not rectified, and all cells such that $\eta_{\alpha }=1$ are rectified. To each such η is associated a unique integer (e.g., $n=\sum_{\alpha =1}^{N_{B}+N_{A}}\eta_{\alpha }2^{\alpha -1}$, η is then the binary coding of *n*). Finally, we set $\Omega^{\left(n\right)}=\Gamma^{\left(n\right)}\times R^{N_{G}}$, where the product with the subspace $R^{N_{G}}$ integrates the states space of RG cells dynamics. They are slaved by B cells and A cells dynamics, but they are not rectified. In this setting, $\Omega^{\left(0\right)}$ is the subset of Ω such that neither B cells nor A cells are rectified; $\Omega^{\left(1\right)}$ the subset of the phase space where only B cell 1 is rectified; $\Omega^{\left(3\right)}$ the subset where only B cells 1,2 are rectified; $\Omega^{\left(2^{N_{B}}\right)}$ the subset where only A cell 1 is rectified and so on.

It is easy to check that the sets $\Omega^{\left(n\right)}$ are disjoint and cover $R^{N}$, and thus make a partition of the phase space. The vector $\vec{R}\left(\vec{X}\right)$ now has the form:$$R_{\alpha }\left(\vec{X}\right)=\left\{\begin{array}{cccc} & \left(1-\eta_{\alpha }\right)\;\left(\;X_{\alpha }-\theta_{B}\;\right),\; & \alpha =1\cdots N_{B};\\ & \left(1-\eta_{\alpha }\right)\;\left(\;X_{\alpha }-\theta_{A}\;\right),\; & \alpha =N_{B}\cdots N_{B}+N_{A};\\ & 0,\; & \alpha =N_{B}+N_{A}+k, & k=1\cdots N_{G}, \end{array}\right.$$
and is piecewise-linear in $\vec{X}$. For $\vec{X}\in \Omega^{\left(n\right)}$, the transformation $T^{-1}.\vec{X}+W.\vec{R}\left(\vec{X}\right)$ can therefore be written $L^{\left(n\right)}.\vec{X}+{\vec{C}}^{\left(n\right)}$, where ${\vec{C}}^{\left(n\right)}$ is the vector with entries:(14)$${\vec{C}}^{\left(n\right)}=\left\{\begin{array}{ccc} -\theta_{B}\;\left(1-\eta_{\alpha }\right)\;\sum_{j}W_{B_{\alpha }}^{A_{j}}, & \; & \alpha =1\cdots N_{B};\\ -\theta_{A}\;\left(1-\eta_{\alpha }\right)\;\sum_{i}W_{A_{\alpha }}^{B_{i}}, & \; & \alpha =N_{B}\cdots N_{B}+N_{A};\\ 0, & \; & \alpha =N_{B}+N_{A}+1,\cdots ,N. \end{array}\right.$$

This is a time-constant vector, coming from the presence of a threshold in rectification (it is zero when $\theta_{A},\theta_{B}=0$), depending on the rectification state of cells, thus depending on the domain $\Omega^{\left(n\right)}$. Rectified cells have zero entries in ${\vec{C}}^{\left(n\right)}$. The matrix:(15)$$L^{\left(n\right)}=\left(\;\begin{array}{cccccc} \; & -\mathrm{diag}{\left[\;\frac{1}{\tau_{B_{i}}}\;\right]}_{i=1\cdots N_{B}} & \; & W_{B_{}}^{A_{}}.D_{A}^{\left(n\right)} & \; & 0_{N_{B}N_{G}}\\ \; & W_{A_{}}^{B_{}}.D_{B}^{\left(n\right)} & \; & -\mathrm{diag}{\left[\;\frac{1}{\tau_{A_{j}}}\;\right]}_{j=1\cdots N_{A}} & \; & 0_{N_{A}N_{G}}\\ \; & W_{G_{}}^{B_{}}.D_{B}^{\left(n\right)} & \; & W_{G_{}}^{A_{}}.D_{A}^{\left(n\right)} & \; & -\mathrm{diag}{\left[\;\frac{1}{\tau_{G_{k}}}\;\right]}_{k=1\cdots N_{G}} \end{array}\;\right),$$
is called the *transport operator in the domain $\Omega^{\left(n\right)}$*. This terminology is further explained in Section 2.1.9, but, in short, $L^{\left(n\right)}$ acts as a flow (or a propagator) characterizing the evolution of a trajectory within $\Omega^{\left(n\right)}$. In Equation (Equation 15), the matrices $D_{B}^{\left(n\right)}=\mathrm{diag}{\left[\;1-\eta_{\alpha }\;\right]}_{\alpha =1\cdots N_{B}}$, $D_{A}^{\left(n\right)}=\mathrm{diag}{\left[\;1-\eta_{\alpha }\;\right]}_{\alpha =N_{B}+1\cdots N_{B}+N_{A}}$ are projecting onto the subspace of non-rectified cells in the domain $\Omega^{\left(n\right)}$. In other words, when the state $\vec{X}$ is in $\Omega^{\left(n\right)}$, a rectified cell α gives a zero contribution to the dynamics of other cells, which corresponds to have a row and column α made of zeros in $D_{A}^{\left(n\right)},D_{B}^{\left(n\right)}$.

The dynamical system (Equation 13) reads now:(16)$$\frac{d\vec{X}}{dt}=L^{\left(n\right)}.\vec{X}+{\vec{F}}^{\left(n\right)}\left(t\right),\;\vec{X}\in \Omega^{\left(n\right)},$$
where we wrote ${\vec{F}}^{\left(n\right)}\left(t\right)={\vec{C}}^{\left(n\right)}+\vec{F}\left(t\right)$. Thanks to the decomposition of the phase space into convex sub-domains $\Omega^{\left(n\right)}$, (Equation 16) is now linear. This technique of phase space decomposition is classical and has been used in domains such as ergodic theory and billiards, self-organized criticality [59,60] or neurosciences [61,62,63,64]. See especially the recent paper by A. Rajakumar et al. [65], very much in the spirit of the present model.

#### 2.1.7. Spectra and Fixed Points

It is important to consider in detail the spectrum of $L^{\left(n\right)}$ for further studies. (Anothe approach consists of considering the Schur decomposition instead of the diagonalisation [65,66,67].) We note $\lambda_{\beta }^{\left(n\right)},\beta =1\cdots N$, the eigenvalues of $L^{\left(n\right)}$, and its right eigenvectors are noted, $P_{\beta }^{\left(n\right)}$. These vectors are the columns of the matrix $P_{}^{\left(n\right)}$ transforming $L^{\left(n\right)}$ in diagonal form (assuming it is diagonalizable). ${\left(\;P_{}^{\left(n\right)}\;\right)}^{-1}$ is the inverse matrix. Its rows are the left eigenvectors of $L^{\left(n\right)}$.

As $D_{A}^{\left(n\right)},D_{B}^{\left(n\right)}$ are projection matrices, it is easy to see, from the form (Equation 15), that a rectified cell generates an eigenvalue $-\frac{1}{\tau_{\alpha }}$ and an eigenvector ${\vec{e}}_{\alpha }$, the canonical basis vector of $R^{N}$ in the direction α. The non-rectified cells span a subspace of $R^{N}$ and the projection of $L^{\left(n\right)}$ on this subspace has a spectrum depending on the connectivity matrices $W_{A_{}}^{B_{}},W_{B_{}}^{A_{}}$ and on other parameters like characteristic times.

The corresponding eigenvalues $\lambda_{\beta }^{\left(n\right)},\beta =1\cdots N$ can be real or complex, with a positive or a negative real part. In the case where $W_{A_{}}^{B_{}}$ and $W_{B_{}}^{A_{}}$ commute, it is actually possible to explicitly compute the eigenvalues and the eigenvectors and obtain conditions for stability (all eigenvalues have real negative part) and real/complex eigenvalues [45]. If we further assume that $W_{A_{}}^{B_{}}$ and $W_{B_{}}^{A_{}}$ have no zero eigenvalues, the sign constraints on these matrices imply that $L^{\left(n\right)}$ is invertible for all *n*. This is what we are going to assume from now.

It follows that, in the absence of external stimulus ($\vec{F}\left(t\right)=\vec{0}$), Equation (Equation 16) has, for each *n*, a unique fixed point ${\vec{X}}^{\left(n\right)}=-{\left(\;L^{\left(n\right)}\;\right)}^{-1}.{\vec{C}}^{\left(n\right)}$. Note, however, that this point *may not be* in $\Omega^{\left(n\right)}$. This is a typical situation for piecewise linear dynamical systems (like Iterated Function Systems [68,69,70]) where dynamics can have complex attractors even if maps are linear (and contracting) into sub-domains of the phase space. The simplest non trivial case is when dynamics generates a periodic orbit, but more complex attractors (fractal sets) can be obtained. Here, it is reasonable to assume at least that cells at rest are not rectified. Mathematically, this means that the fixed point of $L^{\left(0\right)}$, ${\vec{X}}^{*}=-{\left(\;L^{\left(0\right)}\;\right)}^{-1}.{\vec{C}}^{\left(0\right)}$, belongs to $\Omega^{\left(0\right)}$, and this is what we are going to assume for now. This imposes a set of constraints linking synaptic weights and thresholds. A simple assumption consists of having vanishing thresholds $\theta_{A}=\theta_{B}=0$, in which case the rest state is $\vec{0}$. We will also assume that ${\vec{X}}^{*}$ is stable (eigenvalues of $L^{\left(0\right)}$ have a negative real part), which imposes additional assumptions on synaptic weights and cell integration times. On biophysical grounds, it means that the rest state is stable to small perturbations, like noise. Because rectified cells produce stable eigenvalues, the following holds. Taking an initial condition in any domain $\Omega^{\left(n\right)}$, spontaneous dynamics (without stimulus) eventually drive the trajectory back to $\Omega^{\left(0\right)}$ and, then, to the rest state. This is further commented below (Section 2.1.9, remark 2).

#### 2.1.8. Extensions: Gain Control and Gap Junctions

##### Gap Junctions

Electric synapses, e.g., between B cells and A cells, play an important role in the retina, for example in the rod–cone pathway [17]. We consider here passive gap junctions corresponding to electric synapses with a constant conductance (in contrast to conductances depending on variables such as light illumination, see https://webvision.med.utah.edu/book/part-iii-retinal-circuits/myriad-roles-for-gap-junctions-in-retinal-circuits/ (acessed on 22 November 2021)). Let us consider, for example, a gap junction between B cell *i* and A cell *j*. We note $g_{B_{i}}^{A_{j}}\geq 0$ the electric conductance from *j* to *i* (with $g_{B_{i}}^{A_{j}}=0$ if there is no electric connection between the two cells). As gap junctions are symmetric, $g_{B_{i}}^{A_{j}}=g_{A_{j}}^{B_{i}}$. We also note $C_{B_{i}}$ the membrane capacitance of B cell *i* and $C_{A_{j}}$ the membrane capacitance of A cell *j* and introduce the notation $G_{B_{i}}^{A_{j}}=\frac{g_{B_{i}}^{A_{j}}}{C_{B_{i}}}$, $G_{A_{j}}^{B_{i}}=\frac{g_{A_{j}}^{B_{i}}}{C_{A_{j}}}$. Remark therefore that $G_{B_{i}}^{A_{j}}=G_{A_{j}}^{B_{i}}$ if and only if B cell *i* and A cell *j* have the same capacitance. The electric synapse generates a (signed) current $-G_{B_{i}}^{A_{j}}\;\left(\;V_{B_{i}}-V_{A_{j}}\;\right)$ feeding B cell *i* and a current $-G_{A_{j}}^{B_{i}}\;\left(\;V_{A_{j}}-V_{B_{i}}\;\right)$ feeding A cell *j*. Note that, in contrast to chemical synapses, voltages are not rectified, ionic currents are simply following the gradients of electric potentials and can, therefore, go both ways.

The presence of electric synapses between B cells and A cells modifies therefore Equation (Equation 9) as:$$\left\{\begin{array}{cccc} \frac{dV_{B_{i}}}{dt} & = & -\frac{1}{\tau_{B_{i}}^{\prime }}V_{B_{i}}+\sum_{j=1}^{N_{A}}\left[\;W_{B_{i}}^{A_{j}}\;N_{A}\left(\;V_{A_{j}}\;\right)+G_{B_{i}}^{A_{j}}\;V_{A_{j}}\;\right]+F_{B_{i}}\left(t\right), & i=1\cdots N_{B};\\ \frac{dV_{A_{j}}}{dt} & = & -\frac{1}{\tau_{A_{j}}^{\prime }}V_{A_{j}}+\sum_{i=1}^{N_{B}}\left[\;W_{A_{j}}^{B_{i}}\;N_{B}\left(\;V_{B_{i}}\;\right)+G_{A_{j}}^{B_{i}}\;V_{B_{i}}\;\right], & j=1\cdots N_{A}; \end{array}\right.$$
where $\frac{1}{\tau_{B_{i}}^{\prime }}=\frac{1}{\tau_{B}}+\frac{1}{C_{B_{i}}}\;\sum_{j=1}^{N_{A}}g_{B_{i}}^{A_{j}}$, $\frac{1}{\tau_{A_{j}}^{\prime }}=\frac{1}{\tau_{A}}+\frac{1}{C_{A_{j}}}\;\sum_{i=1}^{N_{B}}g_{A_{j}}^{B_{i}}$ are inverse of characteristic time. Thus, gap junctions have the effect of reducing the characteristic time of cell response (increase their conductance). Gap junctions between A cells and RG cells, or between RG cells, would be implemented the same way.

##### Gain Control

This mechanism plays a prominent role in the nervous system. In short, this is the property that neural systems have to adjust the nonlinear transfer function relating input to output to dynamically span the varying range of incoming stimuli [71]. It has been reported in the retina and invoked in several motion processing features: anticipation, alert response to motion onset and motion reversal [42,44]. In particular, B cells have gain control. Here, this is a desensitization when activated by a steady illumination [72], mediated by a rise in intracellular calcium $Ca^{2+}$, at the origin of a feedback inhibition thus preventing prolonged signalling of the ON B cell [44,73]. It can be modelled as follows [42,44,45]. Each B cell has a dimensionless activity variable $a_{B_{i}}$ obeying the differential equation:(17)$$\frac{da_{B_{i}}}{dt}=-\frac{a_{B_{i}}}{\tau_{a}}+h_{B}\;N_{B}\left(\;V_{B_{i}}\;\right).$$

The gain function is a strongly nonlinear function, almost step-wise, of the form:(18)$$G_{B}\left(a_{B_{i}}\right)=\left\{\begin{array}{cc} 0, & \mathrm{if}\;a_{B_{i}}\leq 0;\\ \frac{1}{1+a_{B_{i}}^{6}}, & \mathrm{else}. \end{array}\right.$$

The effect of gain control acts at the level of synaptic transmission from B cells to A cells, where the rectification term $N_{B}\left(\;V_{B_{i}}\;\right)$ is replaced by $N_{B}\left(\;V_{B_{i}}\;\right)\;G_{B}\left(a_{B_{i}}\right)$. That is the equation ruling the A cell *j*’s voltage reads now:$$\frac{dV_{A_{j}}}{dt}=-\frac{1}{\tau_{A}}V_{A_{j}}+\sum_{i=1}^{N_{B}}W_{A_{j}}^{B_{i}}\;N_{B}\left(\;V_{B_{i}}\;\right)\;G_{B}\left(a_{B_{i}}\right).$$

It has the following meaning. When the voltage of B cell *i* increases, its activity $a_{B_{i}}$ increases as well, up to a range where gain control takes place. When $a_{B_{i}}$ becomes too large, $G_{B}\left(a_{B_{i}}\right)$ drops down, thereby reducing the action of B cell *i* on A cell *j*. As mentioned earlier, this is a way to rectify voltages from above.

Gain control has also been reported for (OFF) RG cells [42,44] and shape their firing rate. Gain control at the level of B cells and RG cells induces retinal anticipation. When combined with A cells’ lateral connectivity or gap junctions’ connectivity, it results in a wave of activity ahead of the propagating stimulus (e.g., a moving bar) for specific ranges of parameters (characteristic times of cells response, weight intensities) as studied in [45].

##### Piecewise Linear System with Gain Control and Gap Junctions

Here, we want to expose how the piecewise linear formalism developed above can be applied in the case of gain control and gap junctions. Note that gap junctions actually do not pose any problem from this perspective because they add linear contributions. In the presence of gain control and gap junctions, the dynamical system (Equation 9) becomes:(19)$$\left\{\begin{array}{cccc} \frac{dV_{B_{i}}}{dt} & = & -\frac{1}{\tau_{B_{i}}^{\prime }}V_{B_{i}}+\sum_{j=1}^{N_{A}}\left[\;W_{B_{i}}^{A_{j}}\;N_{A}\left(\;V_{A_{j}}\;\right)+G_{B_{i}}^{A_{j}}\;V_{A_{j}}\;\right]+F_{B_{i}}\left(t\right), & i=1\cdots N_{B};\\ \; & \; & \\ \frac{dV_{A_{j}}}{dt} & = & -\frac{1}{\tau_{A_{j}}^{\prime }}V_{A_{j}}+\sum_{i=1}^{N_{B}}\left[\;W_{A_{j}}^{B_{i}}\;N_{B}\left(\;V_{B_{i}}\;\right)\;G_{B}\left(a_{B_{i}}\right)+G_{A_{j}}^{B_{i}}\;V_{B_{i}}\;\right], & j=1\cdots N_{A};\\ \; & \; & \\ \frac{dV_{G_{k}}}{dt} & = & -\frac{1}{\tau_{G}}V_{G_{k}}+\sum_{i=1}^{N_{B}}W_{G_{k}}^{B_{i}}N_{B}\left(V_{B_{i}}\right)\;G_{B}\left(a_{B_{i}}\right)+\sum_{j=1}^{N_{A}}W_{G_{k}}^{A_{j}}N_{A}\left(V_{A_{j}}\right), & k=1\cdots N_{G};\\ \; & \; & \\ \frac{da_{B_{i}}}{dt} & = & -\frac{a_{B_{i}}}{\tau_{a}}+h_{B}\;N_{B}\left(\;V_{B_{i}}\;\right), & i=1\cdots N_{B}; \end{array}\right.$$

We do not know about experimental evidence of gain control in A cells. This is why A cells are not gain controlled in (Equation 19), but the extension is straightforward. RG cells are gain controlled at the level of their firing rate (see [44]).

To make (Equation 19) a piecewise linear dynamical system, the trick is to replace the function (Equation 18) with a step function where $G_{B}\left(a_{B_{i}}\right)=1$ if $a_{B_{i}}\in \left[\;0,\theta_{a}\;\right]$, where $\theta_{a}$ is a threshold (typically $\frac{2}{3}$ coming from a linear interpolation of (Equation 18); see [45]) and $G_{B}\left(a_{B_{i}}\right)=0$, otherwise. In addition to the rectification variables $\eta_{\alpha }$, we introduce gain control variables $g_{\alpha }=1$ if $a_{B_{i}}\in \left[\;0,\theta_{a}\;\right]$ and $g_{\alpha }=0$ otherwise, $\alpha =1\cdots N_{B}$. The definition of the domains $\Omega^{\left(n\right)}$ extends easily in this context by partitioning $R^{N+N_{B}}$ into sub-domains taking the product of the voltage partition $\left\{\;]-\infty ,\theta_{B}],]\theta_{B},+\infty ]\;\right\}$ with the activity partition $\left\{\;]-\infty ,\theta_{a}],]\theta_{a},+\infty ]\;\right\}$. The transport operator generalizes to: (20)$$L^{\left(n\right)}=\left(\;\begin{array}{cccccccc} \; & -\mathrm{diag}{\left[\;\frac{1}{\tau_{B_{i}}^{\prime }}\;\right]}_{i=1\cdots N_{B}} & \; & W_{B_{}}^{A_{}}.D_{A}^{\left(n\right)}+G_{B_{}}^{A_{}} & \; & 0_{N_{B}N_{G}} & \; & 0_{N_{B}N_{B}}\\ \; & W_{A_{}}^{B_{}}.D_{B}^{{}^{\prime }\left(n\right)}+G_{A_{}}^{B_{}} & \; & -\mathrm{diag}{\left[\;\frac{1}{\tau_{A_{j}}^{\prime }}\;\right]}_{j=1\cdots N_{A}} & \; & 0_{N_{A}N_{G}} & \; & 0_{N_{A}N_{B}}\\ \; & W_{G_{}}^{B_{}}.D_{B}^{{}^{\prime }\left(n\right)} & \; & W_{G_{}}^{A_{}}.D_{A}^{\left(n\right)} & \; & -\mathrm{diag}{\left[\;\frac{1}{\tau_{G_{k}}}\;\right]}_{k=1\cdots N_{G}} & \; & 0_{N_{G}N_{B}}\\ \; & h_{B}\;I_{N_{B}\;N_{B}} & \; & 0_{N_{B}N_{A}} & \; & 0_{N_{B}N_{G}} & \; & -\mathrm{diag}{\left[\;\frac{1}{\tau_{a}}\;\right]}_{i=1\cdots N_{B}} \end{array}\;\right),$$
where $I_{N_{B}\;N_{B}}$ is the $N_{B}$-dimensional identity matrix and $D_{B}^{{}^{\prime }\left(n\right)}=\mathrm{diag}{\left[\;\left(1-\eta_{\alpha }\right)\;g_{\alpha }\;\right]}_{\alpha =1\cdots N_{B}}$. Thus, $D_{B}^{{}^{\prime }\left(n\right)}$ has zero entries whenever a B cell is either rectified ($\eta_{\alpha }=1$) or gain controlled ($g_{\alpha }=0$) leading to a projection on the subspace of B cells which are neither rectified nor gain controlled. Extending the phase space with activity variables corresponds to adding $N_{B}$ eigenvalues $-\frac{1}{\tau_{a}}$ to the spectrum. The corresponding eigenvectors are generalized eigenvectors though because the activities variables add a Jordan block to the matrix [45].

#### 2.1.9. Solutions

We now consider the general situation where dynamics is in the rest state at times t<0, and, from time t=0 on, the stimulus S(x,y,t) is applied, resulting in a non-stationary drive $\vec{F}\left(t\right)$. In general, the stimulus is applied over a finite time. After this, the system eventually returns to rest. Under this stimulation, the trajectory ${\left\{\;\vec{X}\left(t\right)\;\right\}}_{t\geq 0}$ is going to cross a sequence of domains $\Omega^{\left(n_{k}\right)}$, k=1,⋯, with $n_{1}=0$, entirely determined by the stimulus and the network characteristics. Call $t_{-}^{\left(n_{k+1}\right)}$ the time where the trajectory enters the domain $\Omega^{\left(n_{k+1}\right)}$ and $t_{+}^{\left(n_{k+1}\right)}$ the time where it gets out. Note that $t_{-}^{\left(n_{k+1}\right)}=t_{+}^{\left(n_{k}\right)}$. By direct integration of Equation (Equation 16), we have:(21)$$\vec{X}\left(t\right)=e^{L^{\left(n_{k+1}\right)}\left(t-t_{-}^{\left(n_{k+1}\right)}\right)}.\vec{X}\left(t_{-}^{\left(n_{k+1}\right)}\right)\;+\;\int_{t_{-}^{\left(n_{k+1}\right)}}^{t}e^{L^{\left(n_{k+1}\right)}\left(t-s\right)}.{\vec{F}}^{\left(n_{k+1}\right)}\left(s\right)\;ds,\;t\in \left[\;t_{-}^{\left(n_{k+1}\right)},t_{+}^{\left(n_{k+1}\right)}\;\right],$$
where $\vec{X}\left(t_{-}^{\left(n_{k+1}\right)}\right)$, corresponding to the state of $\vec{X}$ when entering $\Omega^{\left(n_{k+1}\right)}$, is given by the integration of the past trajectory and can be computer explicitly. This is:(22)$$\vec{X}\left(t_{-}^{\left(n_{k+1}\right)}\right)=\vec{X}\left(t_{+}^{\left(n_{k}\right)}\right)=\sum_{m=0}^{k}H_{m}^{k}\;{\vec{\Phi }}_{m},$$
where $H_{m}^{k}$ is a sequence of matrices satisfying:(23)$$H_{k}^{k}=I_{N};\;H_{m}^{k}=H_{k-1}^{k}\;H_{m}^{k-1};\;H_{k-1}^{k}=e^{L^{\left(n_{k}\right)}\left(t_{+}^{\left(n_{k}\right)}-t_{+}^{\left(n_{k-1}\right)}\right)},$$
where $I_{N}$ is the identity matrix of dimension *N*. The matrix $H_{m}^{k}$ transports the flow from the exit point of $\Omega^{\left(n_{m}\right)}$ to the exit point of $\Omega^{\left(n_{k}\right)}$. The vectors ${\vec{\Phi }}_{m}$ are defined by:(24)$${\vec{\Phi }}_{0}=\vec{X}\left(0\right);\;{\vec{\Phi }}_{m}=\int_{t_{-}^{\left(n_{m}\right)}}^{t_{+}^{\left(n_{m}\right)}}e^{L^{\left(m\right)}\left(t_{+}^{\left(n_{m}\right)}-s\right)}.{\vec{F}}^{\left(m\right)}\left(s\right)\;ds.$$

The proof of (Equation 22) is easily done by recurrence.

#### 2.1.10. Remarks

Let us now make some remarks on the structure of these solutions.

The interpretation of (Equation 22) is the following. Starting from an initial condition $\vec{X}\left(0\right)\in \Omega^{\left(n_{1}\right)}$, the dynamics (Equation 19) is integrated up to the possible time $t=t_{-}^{\left(n_{2}\right)}=t_{+}^{\left(n_{1}\right)}$ when $\vec{X}\left(t\right)$ gets out of $\Omega^{\left(n_{1}\right)}$ and enters a new domain $\Omega^{\left(n_{2}\right)}$. This arises if, during the time evolution of the system, some cells get rectified (or gain controlled) at time *t*. Then, there is a drastic change in time evolution because rectified cells do not participate in dynamics anymore. The value of the state vector at this time is $\vec{X}\left(t_{+}^{\left(n_{1}\right)}\right)=e^{L^{\left(n_{1}\right)}\left(t_{+}^{\left(n_{1}\right)}-t_{-}^{\left(n_{1}\right)}\right)}.\vec{X}\left(0\right)+\int_{t_{-}^{\left(n_{1}\right)}}^{t_{+}^{\left(n_{1}\right)}}e^{L^{\left(n_{1}\right)}\left(t-s\right)}.{\vec{F}}^{\left(n_{1}\right)}\left(s\right)\;ds$ which can be written $\vec{X}\left(t_{+}^{\left(n_{1}\right)}\right)=H_{0}^{1}.{\vec{\Phi }}_{0}+H_{1}^{1}.{\vec{\Phi }}_{1}=\sum_{m=0}^{1}H_{m}^{1}\;{\vec{\Phi }}_{m}$ using $t_{-}^{\left(n_{1}\right)}=0$. The system is now in the domain $\Omega^{\left(n_{2}\right)}$ and follows its evolution until the (possible) time $t_{-}^{\left(n_{3}\right)}=t_{+}^{\left(n_{2}\right)}$ when some new cells are rectified or some rectified cells become non-rectified. The system enters a new domain $\Omega^{\left(n_{3}\right)}$ and so on. In general, the state at the entrance of domain $\Omega^{\left(n_{k+1}\right)}$ is given by (Equation 22). This is a linear combination of terms $H_{m}^{k}\;{\vec{\Phi }}_{m}$ where ${\vec{\Phi }}_{m}$ (Equation (Equation 24)) integrates the stimulus contribution from the entrance time into domain $\Omega^{\left(n_{m}\right)}$ up to the exit time of this domain and $H_{m}^{k}$ transports the state from the exit point of $\Omega^{\left(n_{m}\right)}$ to the exit point of $\Omega^{\left(n_{k}\right)}$.In the definition of $H_{m}^{k}$, the operators $L_{n_{k}}$ do not commute in general.Eigenvalues of some $H_{m}^{k}$ can have a positive real part leading to an exponential increase along the corresponding eigendirection. This means that some cell voltage increases exponentially in absolute value. However, when voltage becomes too large, voltage rectification (or gain control) takes place, corresponding to the trajectory entering a new continuity domain. Here, unstable cells do not contribute to dynamics anymore, which are projected on the subspace of non-rectified cells. This has the effect of transforming unstable eigenvalues into stable ones preventing the trajectories $\vec{X}\left(t\right)$ from diverging. Actually, the spectrum of $H_{m}^{k}$, controlling stability, resembles the Lyapunov spectrum in ergodic theory [74], with two main differences. First, we are simply considering product of matrices without multiplying by the adjoin so that eigenvalues can be complex. Second, we are not assuming stationarity and the existence of an invariant measure. Instead, the product $H_{m}^{k}$ is constrained by the non-stationary stimulus and dynamical system parameters which fixes the sequence of times $n_{k}$s.Rectification induces a weak form of nonlinearity where e.g., the contraction/expansion in the phase space depends on the domain $\Omega^{\left(n_{k}\right)}$ (whereas, in a differentiable nonlinear system, it would depend on the point in the phase space). This has deep consequences on cells response, as mentioned in the Results section.

### 2.2. Spike Statistics

As pointed out in the Introduction, it might be helpful to propose a mathematical setting taking into account non-stationarity and potentially long memory in spike trains’ probabilities. Such a setting has existed for a long time but has not been applied to spike train statistics until recently. It is inherited from statistical physics on one hand [75] and on extensions of Markov chains to unbounded memory on the other hand [76]. The material briefly sketched here has been published in [46,47,48,49,50,51].

#### 2.2.1. Mathematical Setting for Spike Trains

Neuron variables such as membrane potential or ionic currents are described by continuous-time equations. In contrast, spikes resulting from the experimental observation are discrete events, binned with a certain time resolution δ, say on the order of a millisecond. We consider a network of *N* spiking neurons, labelled with an index k=1⋯N. We define a spike variable $\omega_{k}\left(n\right)=1$ if neuron *k* has emitted a spike in the time interval [nδ,(n+1)δ[, and $\omega_{k}\left(n\right)=0$ otherwise. We denote by $\omega \left(n\right)={\left[\;\omega_{k}\left(n\right)\;\right]}_{k=1}^{N}$ the spike-state of the entire network at time *n*, which we call a *spiking pattern*. A *spike block* denoted by $\omega_{m}^{n}$, n≥m, is the sequence of spike patterns ω(m),ω(m+1)⋯ω(n). The range of a block $\omega_{m}^{n}$ is n−m+1, the number of time steps from *m* to *n*. We call a spike train an infinite sequence of spikes both in the past and in the future, and, to simplify notations, we note a spike train ω (instead of $\omega_{-\infty }^{+\infty }$). Of course, on operational grounds, spike trains are finite, but it is mathematically more convenient to work on a space of bi-infinite spike sequences.

#### 2.2.2. Mathematical Setting for Spiking Probabilities

We now consider a family of transition probabilities of the form $P_{n}\left[\;\omega \left(n\right)\;\left|\;\omega_{-\infty }^{n-1}\right.\;\right]$, which represents the probability that, at time *n*, one observes the spiking pattern ω(n) given the network spike history, extending to an infinite past. This is an extension of Markov chains where probabilities have the form $P_{n}\left[\;\omega \left(n\right)\;\left|\;\omega_{n-D}^{n}\right.\;\right]$, where *D* is the memory depth of the Markov chain. Letting the memory be possibly infinite corresponds to a situation where one cannot precisely fix the memory depth necessary to characterize the probability of a spike pattern given the past spike history. An example of a model requiring this context is presented in Section 2.2.3 below. Having infinite memory imposes mathematical constraints on the memory decay that has to be sufficiently fast (typically, exponential) so that the situation is close to Markov chains. In addition to the model presented below, neural models with infinite memories have been considered by several authors such as E. Loecherbach and A. Galves [77]. A few remarks about this form of probability:We do not assume stationarity. $P_{n}$ may depend explicitly on time. This is actually the reason why we have an index *n*. A time translation invariant probability will simply be written P.For such probabilities to be well defined and useful, one needs to make assumptions on their structure. Beyond technical assumptions such as measurability, summability, non-nullness and continuity [78,79], the most important assumption here is that the dependence in the past (memory) decays fast enough, typically, exponentially, so that, even if this chain has infinite memory, it is very close to Markov.As one can associate to Markov chains an equilibrium probability (under conditions actually quite more general than detailed-balance), the system of transition probabilities ${\left\{P_{n}\right\}}_{n\in Z}$ also admits, under the mathematical conditions sketched in the item 2 above, an equivalent notion called “chains with complete connections” or a “chain with unbounded memory” [76].These distributions are formally (left-sided) Gibbs distributions where the Gibbs potential is $\Phi \left(n,\omega \right)\overset{\mathrm{def}}{=}\log P_{n}\left[\;\omega \left(n\right)\;\left|\;\omega_{-\infty }^{n-1}\right.\;\right]$ (the non-nullness assumption imposes that $P_{n}\left[\;\omega \left(n\right)\;\left|\;\omega_{-\infty }^{n-1}\right.\;\right]>0$). This establishes a formal link to statistical physics. In particular, when the chain is stationary, expanding the potential in product of spikes events up to the second order, one recovers the maximum entropy models used in the literature of spike trains analysis, including the so-called Ising model [22,48,80,81]. However, the chains we consider are not necessarily stationary.

#### 2.2.3. A Model of Effective Interactions between RG Cells

The visual cortex has no clue on which biophysical processes are taking place in the retina. All the visual information it receives is encoded in spike trains. This leads to the idea of proposing models of a spiking RG cells network where dynamics of RG cells voltage are only constrained by RG cells’ spikes history. Here, one assumes that RG cells dynamics are controlled by the interactions with hidden layers, for example, the B cells–A cells layers in the model (Equation 13), in a situation where an observer is just recording the spikes emitted by RG cells, while having no clue of the dynamics in the upper layers. These hidden layers result in providing *effective interactions* between RG cells that one can interpolate by fitting the statistics. The idea is then to construct a dynamical model where the spiking of an RG cell depends on the spike history emitted by the network, with virtual interactions that mimic hidden causal effects [82]. This strategy leads us to propose the model presented in the next paragraph. The advantage of this approach is that one can explicitly write the transition probabilities $P_{n}\left[\;\omega \left(n\right)\;\left|\;\omega_{-\infty }^{n-1}\right.\;\right]>0$ and infer, from this, a linear response formula telling us how statistical quantities such as firing rates, but also spike correlations are modified by a time dependent stimulus. These results are presented in the “Results” Section 2.2.3.

The model is inspired from the generalized Integrate and Fire model (gIF) proposed by Rudolph and Destexhe [83] and generalizes the Leaky-Integrate and Fire (LIF) model [84,85]. We have *N* neurons (say RG cells) characterized by their voltage $V_{k},k=1\cdots N$. One fixes a voltage threshold θ such that, whenever $V_{k}\left(t\right)=\theta$, a spike is emitted by neuron *k* at time *t*, and is reset to a reset value (typically, $V_{reset}=0$). Below θ, the dynamics of voltage (sub-threshold dynamics) are governed by Equation (Equation 27) below.

In the LIF model, synaptic conductances are constant. In the gIF model, in contrast, the synaptic conductance $g_{kj}$ between the pre-synaptic neuron *j* and the post-synaptic neuron *k* depends on spike history as:(25)$$g_{kj}\left(t,\omega \right)=G_{kj}\;\alpha_{kj}\left(t,\omega \right),$$
where:(26)$$\alpha_{kj}\left(t,\omega \right)=\sum_{n=-\infty }^{t}\alpha_{kj}\left(t-n\right)\;\omega_{j}\left(n\right).$$

The notation $g_{kj}\left(t,\omega \right)$ means that function $g_{kj}$ depends on spikes occurring before time *t*. $G_{kj}\geq 0$ is the maximal conductance between *j* and *k*. It is zero when there is no synaptic connection between neurons *j* and *k*. In (Equation 26), the function $\alpha_{kj}\left(t\right)$, called α-kernel, summarizes the complex dynamical process underlying the generation of a post-synaptic potential after the emission of a pre-synaptic spike [86]. It has the typical form $\alpha_{kj}\left(t\right)=P\left(t\right)e^{-\frac{t}{\tau_{kj}}}H\left(t\right)$ where P(t) is a polynomial in time and H(t) is the Heaviside function. What matters on mathematical grounds is the exponential tail of $\alpha_{kj}\left(t\right)$ [46]. The function $\alpha_{kj}\left(t,\omega \right)$ depends on the spike history preceding *t*. It records the spikes emitted by the pre-synaptic neuron *j* before *t*, corresponding to $\omega_{j}\left(n\right)=1$ and adds up a contribution $\alpha_{kj}\left(t-n\right)$ to the post synaptic conductance from pre-synaptic neuron *j* to post-synaptic neuron *k*.

Now, the gIF dynamics reads [46,47,62]:$$C_{k}\frac{dV_{k}}{dt}+g_{L}\left(V_{k}-E_{L}\right)+\sum_{j}g_{kj}\left(t,\omega \right)\left(V_{k}-E_{j}\right)=S_{k}\left(t\right)+\sigma_{B}\xi_{k}\left(t\right),\;\mathrm{if}\;V_{k}\left(t\right)<\theta ,$$
where $g_{L},E_{L}$ are respectively the leak conductance and the leak reversal potential, $E_{j}$ the reversal potential characterizing the synaptic transmission between *j* and *k*. Finally, $\xi_{k}\left(t\right)$ is a white noise, introducing stochasticity in dynamics. Its intensity is $\sigma_{B}$.

Setting $W_{kj}=G_{kj}E_{j}$, $i_{k}\left(t,\omega \right)=g_{L}E_{L}+\sum_{j}W_{kj}\alpha_{kj}\left(t,\omega \right)+S_{k}\left(t\right)+\sigma_{B}\xi_{k}\left(t\right)$, $g_{k}\left(t,\omega \right)=g_{L}+\sum_{j=1}^{N}g_{kj}\left(t,\omega \right)$, one can finally write the gIF dynamics in the form:(27)$$C_{k}\;\frac{dV_{k}}{dt}+g_{k}\left(\;t,\omega \;\right)V_{k}=i_{k}\left(t,\omega \right),\;ifV_{k}\left(t\right)<\theta ,$$
where $i_{k}\left(t,\omega \right)$ depends on the network spike history via $\alpha_{kj}\left(t,\omega \right)$, on the stimulus, and contains a stochastic term. As the reversal potential $E_{j}$ can be positive or negative, the synaptic weights $W_{kj}$ define an oriented and signed graph, whose vertices are the neurons. These weights are what we call effective interactions.

What makes the gIF model very rich is that it proposes a biophysically grounded way to construct a dynamical system where the variables (here, voltages) are constrained by the only information of spike train history. The price to pay is that dynamics actually depend on the *whole spike history*, which is potentially infinite. Actually, the gIF model has an infinite memory. This is essentially because the conductance depends on the whole history, and, contrarily to voltages, is not reset when the neuron fires. Nevertheless, the exponential decay in the alpha profile actually ensures the existence (and uniqueness) of transition probabilities of the form $P_{n}\left[\;\omega \left(n\right)\;\left|\;\omega_{-\infty }^{n-1}\right.\;\right]$ [47,48,49,50].

Note that the integration of (Equation 27) does not only require the knowledge of voltages $V_{k}$, stimulus and noise at time *t*. It requires, in addition, the knowledge of the spike train ω emitted by the network before *t*. In this sense, this is not a classical dynamical system. Nevertheless, Equation (Equation 27) can be explicitly integrated [47,51].

## 3. Results

### 3.1. How Could Lateral A Cells Connectivity Shape the Receptive Field of a Ganglion Cell?

The response of an RG cell to visual stimuli is shaped by the retina structure depicted in Figure 1. Here, with the model introduced in Section 2.1, we would like to characterize the respective effects of the stimulus and of the network connectivity, especially A cells, and understand under which condition can the conjugated effect of network dynamics and stimulus be represented by a convolution of the form (Equation 1) where the kernel $K_{G_{\alpha }}$ is *intrinsic* to the cell, i.e., does not depend on the stimulus?

#### 3.1.1. Non-Rectified Case

The answer is relatively easy when no rectification takes place, i.e., when the trajectory of (Equation 13) stays in the domain $\Omega^{\left(0\right)}$ (see Section 2.1.6 for the definition). Indeed, in this case, evolution is ruled by Equation (Equation 21) which holds from the initial time $t=t_{0}$, where the stimulus starts to be applied, to the current time *t*. Actually, we can consider that $t_{0}$ starts far in the past and let it tend to −∞. This corresponds to considering that the stimulus is applied on a time scale quite longer than the characteristic times in the problem (i.e., the inverse of the real part of eigenvalues). Then, Equation (Equation 21) reads $\vec{X}\left(t\right)=\int_{-\infty }^{t}e^{L^{\left(0\right)}\left(t-s\right)}.{\vec{F}}^{\left(0\right)}\left(s\right)\;ds$, which is $\vec{X}\left(t\right)=\left[\;e^{L^{\left(0\right)}}\;\overset{t}{*}{\vec{F}}_{0}\;\right]\left(t\right)$. This equation actually makes sense only if all eigenvalues of $L^{\left(0\right)}$ are stable, as we assumed above. Note also that ${\vec{F}}^{\left(0\right)}={\vec{C}}^{\left(0\right)}+\vec{F}$ where ${\vec{C}}^{\left(0\right)}$ is a constant, depending on thresholds (Equation (Equation 14)) and whose integration in the convolution product gives $-{\left(\;L^{\left(0\right)}\;\right)}^{-1}.{\vec{C}}^{\left(0\right)}={\vec{X}}^{*}$, the base line activity of $\vec{X}\left(t\right)$ without stimulus. We may ignore this constant in the sequel and focus on the time varying part of the response, $\left[\;e^{L^{\left(0\right)}}\;\overset{t}{*}\vec{F}\;\right]\left(t\right)$. As $\vec{F}$ is itself defined in terms of a convolution (Equation (Equation 5)) with the stimulus and its derivative, $\vec{X}\left(t\right)$ is a convolution with the stimulus and its derivative. Here, it is useful to express $\vec{X}\left(t\right)$ in components.

One can then show that [45]:(28)$$X_{\alpha }\left(t\right)=V_{\alpha_{drive}}\left(t\right)\;+E_{\alpha_{net}}^{\left(0\right)}\left(t\right),\;\alpha =1\cdots N,$$
where:(29)$$E_{\alpha_{net}}^{\left(0\right)}\left(t\right)=\sum_{\beta =1}^{N}\sum_{\gamma =1}^{N_{B}}P_{\alpha \beta }^{\left(0\right)}\;{\left(\;P_{\beta \gamma }^{\left(0\right)}\;\right)}^{-1}\;\varpi_{\beta \gamma }^{\left(0\right)}\;\int_{-\infty }^{t}e^{\lambda_{\beta }^{\left(0\right)}\left(t-s\right)}\;V_{\gamma_{drive}}\left(s\right)\;ds,$$
where $\varpi_{\beta \gamma }^{\left(0\right)}=\lambda_{\beta }^{\left(0\right)}+\frac{1}{\tau_{B_{\gamma }}}$. The term $V_{\alpha_{drive}}\left(t\right)$ in Equation (Equation 28) is the stimulus drive and acts only on B cells (it vanishes for $\alpha >N_{B}$). The term (Equation 29) contains the network effects. The drive imposed on B cells impacts A cells via the connectivity and, thereby, have a feedback effect on B cells. In addition, the join activity of B cells and A cells drive the RG cells response ($\alpha >N_{B}+N_{A}$). In particular, this equation allows for computing explicitly the RF of an RG cell.

For this, we introduce the function $e_{\beta }^{\left(0\right)}\left(t\right)\equiv e^{\lambda_{\beta }^{\left(0\right)}\;t}\;H\left(t\right)$ so that $\int_{-\infty }^{t}e^{\lambda_{\beta }^{\left(0\right)}\left(t-s\right)}\;V_{\gamma_{drive}}\left(s\right)\;$$ds\equiv \left[\;e_{\beta }^{\left(0\right)}\overset{t}{*}V_{\gamma_{drive}}\;\right]\left(t\right)$, which according to (Equation 2) is $\left[\;e_{\beta }^{\left(0\right)}\overset{t}{*}K_{B_{\gamma }}\overset{x,y,t}{*}S\;\right]\left(t\right)$. Thus, by identification with (Equation 1), the kernel of RG cell $\alpha =N_{B}+N_{A}+1\cdots N_{G}$ is:(30)$$K_{G_{\alpha }}\left(x,y,t\right)=\sum_{\beta =1}^{N}\sum_{\gamma =1}^{N_{B}}P_{\alpha \beta }^{\left(0\right)}\;{\left(\;P_{\beta \gamma }^{\left(0\right)}\;\right)}^{-1}\;\varpi_{\beta \gamma }^{\left(0\right)}\left[\;e_{\beta }^{\left(0\right)}\overset{t}{*}K_{B_{\gamma }}\;\right].$$

This provides an explicit equation for the kernel of an RG cell, embedded in a network of B cells, A cells, RG cells with dynamics (Equation 13), when no rectification takes place.

#### 3.1.2. Interpretation

The kernel obtained in (Equation 30) is the response of the RG cell to a Dirac pulse corresponding, in experiments, to a brief light (or dark) full-field flash. It can also obtained from a white noise stimulus, corresponding, in experiments, to the so-called Spike Triggered Average (STA) [38,39,40]. It corresponds therefore to the functional definition of the receptive field of RG cells used in experiments. In addition, Equations (Equation 28) and (Equation 29) give us the voltage of *all* cells in the network at time *t* under the influence of a stimulus. Interestingly, thus, these equations allow us to visualize the join evolution of B cells and A cells as well as their action of RG cells. Note that B cells and A cells are difficult to access experimentally. Given a prescribed connectivity (matrices $W_{A_{}}^{B_{}},W_{B_{}}^{A_{}},W_{G_{}}^{B_{}},W_{G_{}}^{A_{}}$), Equation (Equation 28) provides us, therefore, a mathematical insight on the potential, hidden, dynamics of B cells and A cells leading to the experimentally observed response of RG cells. Thus, this gives us possible scenarios characterizing the potential effects of A cells networks on RG cells response. In addition, Equation (Equation 30) also provides the RF for B cells ($\alpha =1\cdots N_{B}$) and A cells ($\alpha =N_{B}+1\cdots N_{B}+N_{A}$). We observe in particular that, in a network, the RF of a B cell is therefore not only what comes from the OPL—the term $V_{\alpha_{drive}}\left(t\right)$—it integrates as well lateral A cells connectivity. This is similar to the center-surround shaping of OPL output due to H cells, but here, we might have different effects, due to the different physiology of A cells.

#### 3.1.3. Space-Time Separability

The RG cell kernel, in general, does not factorise into a product of a function of space and a function of time (separability). Even in the case where the B cells RF is separable, i.e., $K_{B_{\gamma }}\left(x,y,t\right)=K_{B_{S_{\gamma }}}\left(x,y\right)\;K_{B_{T_{\gamma }}}\left(t\right)$ where $K_{B_{S_{\gamma }}}$ is the spatial part, centred at $x_{\gamma },y_{\gamma }$ and $K_{B_{T_{\gamma }}}$ the temporal part, the RG cell kernel reads:(31)$$K_{G_{\alpha }}\left(x,y,t\right)=\sum_{\beta =1}^{N}P_{\alpha \beta }^{\left(0\right)}\;\left(\;\sum_{\gamma =1}^{N_{B}}{\left(\;P_{\beta \gamma }^{\left(0\right)}\;\right)}^{-1}\;\varpi_{\beta \gamma }^{\left(0\right)}\left[\;e_{\beta }^{\left(0\right)}\overset{t}{*}K_{B_{T_{\gamma }}}\;\right]\times K_{B_{S_{\gamma }}}\left(x,y\right)\;\right),$$
and is not separable either. Now, if B cells have the same temporal kernel $K_{B_{T}}$, independent of γ and the same characteristic time $\tau_{B}$, such that $\varpi_{\beta \gamma }^{\left(0\right)}=\lambda_{\beta }^{\left(0\right)}+\frac{1}{\tau_{B}}$ is independent of γ, we can write:(32)$$K_{G_{\alpha }}\left(x,y,t\right)=\sum_{\beta =1}^{N}P_{\alpha \beta }^{\left(0\right)}\;\varpi_{\beta }^{\left(0\right)}\;\left[\;e_{\beta }^{\left(0\right)}\overset{t}{*}K_{B_{T}}\;\right]\;\left(\;\sum_{\gamma =1}^{N_{B}}{\left(\;P_{\beta \gamma }^{\left(0\right)}\;\right)}^{-1}K_{B_{S_{\gamma }}}\left(x,y\right)\;\right).$$

This kernel is not yet strictly separable as the term $\sum_{\gamma =1}^{N_{B}}{\left(\;P_{\beta \gamma }^{\left(0\right)}\;\right)}^{-1}K_{B_{S_{\gamma }}}\left(x,y\right)$ still depends on β, the eigenmode index, via ${\left(\;P_{\beta \gamma }^{\left(0\right)}\;\right)}^{-1}$. Now, the eigenmodes depend on connectivity. In particular, the B cell to RG cell connectivity corresponds to a pooling of B cells located in the vicinity of RG cell α. The simplest case is when there is no lateral connectivity and where each RG cell α is contacted by only B cell with index $\gamma_{\alpha }$ (this implies $N_{B}=N_{G}$). In this case: $P_{\alpha \beta }^{\left(0\right)}=\delta_{\alpha \beta }$, ${\left(\;P_{\beta \gamma }^{\left(0\right)}\;\right)}^{-1}=\delta_{\beta \gamma }$ so that $K_{G_{\alpha }}\left(x,y,t\right)=\varpi_{\alpha }^{\left(0\right)}\;\left[\;e_{\alpha }^{\left(0\right)}\overset{t}{*}K_{B_{T}}\;\right]\;K_{B_{S_{\alpha }}}\left(x,y\right)$ is separable. More generally, pooling implies that $P_{\alpha \beta }^{\left(0\right)}$ and ${\left(\;P_{\beta \gamma }^{\left(0\right)}\;\right)}^{-1}$ are locally spread around α resulting in a spatial part $\sum_{\gamma =1}^{N_{B}}{\left(\;P_{\beta \gamma }^{\left(0\right)}\;\right)}^{-1}K_{B_{S_{\gamma }}}\left(x,y\right)$ depending only on α.

#### 3.1.4. Resonances

The eigenvalues of $L^{\left(0\right)}$ can be complex, going by conjugated pairs. It is actually quite easy to obtain such a situation mathematically, even considering nearest neighbours’ interactions [45]. A straightforward consequence is the existence of preferred time frequencies (resonance) for an RG cell. In other words, applying periodic sequences of brief flashes with a varying frequency, one might observe a peak in the amplitude of the RG cell response, for specific frequencies. This remark could, e.g., explain the “bump” observed in experiments when the retina is submitted to the so-called “Chirp” stimulus [87], a stimulus composed of different phases of flashes stimulation where one varies duration, frequency, and amplitude. In the phase where the amplitude is constant but frequency is varying, some RG cells exhibit a resonance like peak (see e.g., Figure 1b in [87]). Of course, such resonances could also be explained by intrinsic cells’ properties, like ion channel response. The potential effect of lateral A cells connectivity would have to be tested experimentally by, e.g., inhibiting A cells synaptic transmission for RG cells exhibiting resonance peaks.

#### 3.1.5. Stimulus Induced Waves

This is a general fact that networks of coupled units can produce waves. Spontaneous waves are actually reported in the developmental retina, induced, in the so-called stage II and stage III by A cells [88]. They are generated by nonlinear mechanisms and closeness to bifurcations [27]. This is not the type of wave we are dealing with here, though. Instead, we are referring to waves triggered by a moving stimulus, say a moving bar. The idea is that such a stimulus can induce, via A cells connectivity, a wave of connectivity which can be *ahead* of the stimulus, for a certain range of parameters (e.g., synaptic coupling intensity) compatible with physiology. Stimulus induced waves, in advance with respect to the stimulus, have been reported in the visual cortex [89]. They are due to lateral cortical activity and induce cortical anticipation. The mathematical analysis made in [45] suggests that such anticipatory waves could also exist in the retina thanks to A cells’ lateral connectivity, conjugated with nonlinear gain control already known to induce a form of retinal anticipation [42,44].

#### 3.1.6. Stimulus Adaptation

Short term plasticity has been reported in the retina at the synapses between B cells–A cells and A cells–RG cells [43,90]. Note actually that, although most models of plasticity referring to cortical neurons, are considering spiking neurons [91], the physiology of short-term synapse adaptation does not necessarily require spikes and is compatible with inner retinal networks dynamics. The effect of synaptic plasticity can be integrated in the model (Equation 13). It will result in variations of eigenvalues and eigenvectors of the transport operator L with potential changes in dynamics. Although potential and highly relevant phenomena such as bifurcations induced by plasticity would require considering a nonlinear version of (Equation 13) (at least, rectification to avoid exponential instability), we can ask about the simple linear effect of plasticity on the RG cell response. A straightforward potential effect could be frequency adaptation to periodic flashes.

#### 3.1.7. Rectification

Let us now investigate the role of rectification. In the general case, a trajectory crosses several domains, and is characterized by Equation (Equation 21). Starting from the domain $\Omega^{\left(n_{1}\right)}$ (rest state), the state of the network submitted to a stimulus, enters a new domain $\Omega^{\left(n_{2}\right)}$ at time $t_{+}^{\left(n_{2}\right)}$ where some cells are rectified and so on. Can one still define a response formula of type (Equation 1)? This raises several technical difficulties, first because some eigenvalues can be unstable. As we have seen above, this does not lead to an exponential explosion, though precisely rectification prevents cell voltage from diverging. Mathematically, this is expressed by the exit of the trajectory from the domain with a positive eigenvalue and a projection on the subspace spanned by non-rectified cells. Another difficulty also comes from the constants ${\vec{C}}^{\left(n\right)}$ defined in (Equation 14) coming from the threshold in the rectification function. They can be removed by assuming that all thresholds are equal to 0. This is what we are going to do now for the sake of simplicity. One can then define domain-dependent flows $\Phi^{\left(n\right)}\left(\vec{X},t\right)\equiv e^{L^{\left(n\right)}\;t}\;\Theta \left(\;\vec{X}\left(t\right)\in \Omega^{\left(n\right)}\;\right)$, where Θ is the indicator function so that $\left[\;\Phi^{\left(n\right)}\left(\vec{X},.\right)\overset{t}{*}\vec{F}\;\right]\left(t\right)=\sum_{n_{m}=n}\int_{t_{-}^{\left(n_{m}\right)}}^{t_{+}^{\left(n_{m}\right)}}e^{L^{\left(m\right)}\left(t_{+}^{\left(n_{m}\right)}-s\right)}.\vec{F}\left(s\right)\;ds$, where the sum holds on indices $n_{m}$ in the trajectory such that $n_{m}=n$. This allows us to express the recurrence Formula (Equation 22) in terms of a convolution and thereby to express the whole trajectory in terms of a convolution with a transport operator.

However, there are several important differences with the non-rectified case. First, the kernel defined this way *depends on the trajectory*. As the sequence of domains met by the trajectory (and the time where the trajectory enters in these domains) depend on the stimulus, the RF of rectifiable cells *depends now on the stimulus*. Note that the situation would actually be even worse for nonlinear cells. Indeed, the question hidden behind these remarks is: “to what extent the *linear response* assumption defining a RF via a convolution equation such as (Equation 2) is valid”. We will actually come back to a similar question in Section 3.3 for a network of spiking neurons. Linear response essentially requires the perturbation to be “weak enough”, which, in our case, means that cells are not rectified. The formulation in terms of a piecewise linear system allows for extending the notion of RF to rectified cells, but the price to pay is that RF now depends on the stimulus. With respect to biology, this effect would for example mean that cells identified e.g., to be ON with an STA approach, responds differently (e.g., ON-OFF) to a more sophisticated stimulus like the “chirp“ stimulus [87].

In the rectified cases, the eigenvalues $\lambda_{\beta }^{\left(n\right)},\beta =1\cdots N$ and eigenvectors $P_{\beta }^{\left(n\right)}$ depend on the domain, i.e., on the list of rectified cells and are different from the domain $\Omega^{\left(0\right)}$ of the rest state. They actually differ in two ways. First, rectified cells provide eigenvalues $-\frac{1}{\tau_{\beta }}$ and eigenvectors ${\vec{e}}_{\beta }$ so that $P_{\alpha \beta }^{\left(n\right)}=\delta_{\alpha \beta }$ for these cells so that they do not contribute any more to the network response. The second effect is more intricate. Indeed, the mere fact of rectifying one cell, has, in general, the effect of *modifying the whole spectrum and eigenvectors*, with strong effects on the cell response. This can be easily understood. Consider the (not really retinal-realistic) situation where a cell is a hub in a network. Silencing it has in general dramatic effects on the global dynamics of this network.

#### 3.1.8. Conclusions of This Section

In this section, we have given a mathematical answer to the problem 1, level 1, posed in the Introduction. On the basis of a simplified model of B cells–A cells–RG cells interactions, we have produced a formalism allowing us to compute this network response to spatio-temporal stimuli. We have been able to write explicitly the RF of individual RG cells appearing in Equation (Equation 1) where the kernel depends explicitly on lateral connectivity. As we showed, however, the linear response Formula (Equation 1), where the kernel is independent of the stimulus, holds when the stimuli have a weak enough amplitude so that cells are not rectified. As soon as rectification takes place, the convolution form (Equation 1) implies, in general, that the kernel can change with the stimulus. This effect could be observed in experiments if the cell type, characterized via STA, provides a different type of response to other stimuli.

### 3.2. How Could Spatio-Temporal Stimuli Correlations and Retinal Network Dynamics Shape the Spike Train Correlations at the Output of the Retina?

In this section, we extrapolate the previous analysis of the model (Equation 13) to analyse how spike trains emitted by RG cells can be correlated via the network and especially A cells connectivity. We especially want to make mathematical statements on how could A cells *decorrelate* RG cells, as claimed on the basis of experiments [15]. We consider first the non-rectified case and then analyse how rectification can modify correlations.

#### 3.2.1. Voltage Correlations

We first compute the voltage correlations induced by a non-stationary spatio-temporal stimulus in the model (Equation 13). Note that correlations require some notion of probability and, thus, of randomness. Moreover, it is more convenient when such a probability is stationary, while we want here to consider a non-stationary problem. This is not contradictory though. There are two simple (not incompatible) ways to address this point. First, one may consider that the dynamical system (Equation 13) has random initial conditions, drawn with respect to a stationary probability measure. Second, one can add to the dynamics (Equation 13) noise, which is always present in biological systems. We can make the assumption that noise is stationary and that it is Brownian (which is a pure mathematical convenience). In biology, spike correlations are usually obtained by averaging over repeats of the same experiment where a stimulus is presented to the retinal network. This corresponds therefore to averaging over initial conditions in the presence of noise. Here, to make things simpler, we assume that initial conditions are deterministic (the network is in the rest state when the stimulus is applied) and randomness is induced by a Brownian noise.

##### Stimulus Induced Correlations in the Non-Rectified Case

Let us therefore consider a stimulus with the form $S\left(x,y,t\right)=S_{d}\left(x,y,t\right)+\sigma_{S}\;\xi \left(x,y,t\right)$ where $S_{d}\left(x,y,t\right)$ is deterministic and ξ(x,y,t) is a spatio-temporal white noise. $\sigma_{S}$ controls the intensity of this noise. The spatial integration of B cells RF induces then an obvious correlation between B cell voltages. Consider indeed the term $V_{i_{drive}}\left(t\right)$ in Equation (Equation 2) in the presence of this stimulus. Denoting $E\left[\;\;\right]$ the expectation with respect to the Wiener measure, we have $E\left[\;\xi (x,y,t)\;\right]=0$ and $E\left[\;\xi \left(x,y,t\right)\;\xi \left(x^{\prime },y^{\prime },t^{\prime }\right)\;\right]=\delta \left(x-x^{\prime }\right)\;\delta \left(y-y^{\prime }\right)\;\delta \left(t-t^{\prime }\right)$. Then, $E\left[\;V_{i_{drive}}\left(t\right)\;\right]=\left[\;K_{B_{i}}\overset{x,y,t}{*}S_{d}\;\right]\left(t\right)$ and the correlation between drives is: (33)$$\begin{array}{cc} \; & E\left[\;\left(\;V_{i_{drive}}\left(t\right)-E\left[\;V_{i_{drive}}\left(t\right)\;\right]\;\right)\left(\;V_{j_{drive}}\left(t^{\prime }\right)-E\left[\;V_{j_{drive}}\left(t^{\prime }\right)\;\right]\;\right)\;\right]\\ \; & =\sigma_{S}^{2}\;\int_{x=-\infty }^{+\infty }\;\int_{y=-\infty }^{+\infty }\;\int_{s=-\infty }^{t}K_{B_{i}}\left(x-x_{i},y-y_{i},t-s\right)\;K_{B_{j}}\left(x-x_{j},y-y_{j},t^{\prime }-s\right)\;dx\;dy\;ds, \end{array}$$
assuming $t\leq t^{\prime }$ without loss of generality. We recall that $x_{i},y_{i}$ are the coordinates of the center of BCell *i* RF. Equation (Equation 33) expresses that drives are correlated due to the overlap of B cell RFs, a well known result. In particular, correlations decrease with the distance *d* between the two RF centers (like $e^{-d^{2}}$ if RFs are Gaussian).

More generally, the term $F_{B_{i}}\left(t\right)$ in Equation (Equation 5) has mean:$$E\left[\;F_{B_{i}}\left(t\right)\;\right]=\left[\;\left(\;\frac{1}{\tau_{B}}\;K_{B_{i}}+\frac{\partial }{\partial t}K_{B_{i}}\;\right)\overset{x,y,t}{*}S_{d}\;\right]\left(t\right),$$
and correlation:(34)$$C_{F_{ij}}\left(t,t^{\prime }\right)=\sigma_{S}^{2}\;\left(\;\begin{array}{cc} \; & \frac{1}{\tau_{B}^{2}}\;\int_{x=-\infty }^{+\infty }\;\int_{y=-\infty }^{+\infty }\;\int_{s=-\infty }^{t}K_{B_{i}}\left(x-x_{i},y-y_{i},t-s\right)\;K_{B_{j}}\left(x-x_{j},y-y_{j},t^{\prime }-s\right)\;dx\;dy\;ds\\ \; & \\ \; & +\frac{1}{\tau_{B}}\;\int_{x=-\infty }^{+\infty }\;\int_{y=-\infty }^{+\infty }\;\int_{s=-\infty }^{t}K_{B_{i}}\left(x-x_{i},y-y_{i},t-s\right)\;\frac{\partial }{\partial t}K_{B_{j}}\left(x-x_{j},y-y_{j},t^{\prime }-s\right)\;dx\;dy\;ds\\ \; & \\ \; & +\frac{1}{\tau_{B}}\;\int_{x=-\infty }^{+\infty }\;\int_{y=-\infty }^{+\infty }\;\int_{s=-\infty }^{t}\frac{\partial }{\partial t}K_{B_{i}}\left(x-x_{i},y-y_{i},t-s\right)\;K_{B_{j}}\left(x-x_{j},y-y_{j},t^{\prime }-s\right)\;dx\;dy\;ds\\ \; & \\ \; & +\;\int_{x=-\infty }^{+\infty }\;\int_{y=-\infty }^{+\infty }\;\int_{s=-\infty }^{t}\frac{\partial }{\partial t}K_{B_{i}}\left(x-x_{i},y-y_{i},t-s\right)\;\frac{\partial }{\partial t}K_{B_{j}}\left(x-x_{j},y-y_{j},t^{\prime }-s\right)\;dx\;dy\;ds \end{array},\;\right)$$
for $i,j=1\cdots N_{B}$. This implies that the forcing term $\vec{F}$ in (Equation 13) has a N×N time dependent correlation matrix $C_{F}\left(t,t^{\prime }\right)$ with a $N_{B}\times N_{B}$ block corresponding to (Equation 34), and the rest of the matrix has zeros (A cells and RG cells have no direct stimulus drive).

Let us now consider the full dynamics (Equation 16), in the non-rectified case: the trajectory stays in $\Omega^{\left(0\right)}$. Under the stimulus $S_{d}\left(x,y,t\right)+\sigma_{S}\;\xi \left(x,y,t\right)$, $\vec{X}\left(t\right)$ is a stochastic process, with mean:(35)$$E\left[\;\vec{X}\left(t\right)\;\right]=\left[\;e^{L^{\left(0\right)}}\overset{t}{*}E\left[\;{\vec{F}}^{\left(0\right)}\;\right]\;\right],$$
and correlation matrix:(36)$$C_{\vec{X}}\left(t,t^{\prime }\right)=\int_{s=-\infty }^{t}\;\int_{s^{\prime }=-\infty }^{t^{\prime }}\;e^{L^{\left(0\right)}\left(t-s\right)}\;.C_{F}\left(s,s^{\prime }\right).\;e^{{\tilde{L}}^{\left(0\right)}\left(t^{\prime }-s^{\prime }\right)}\;ds\;ds^{\prime }$$
where ${\tilde{L}}^{\left(0\right)}$ is the transpose of $L^{\left(0\right)}$. This is the general form of correlations induced by the network. Note that correlations are stationary (they only depend on $t-t^{\prime }$). This does not hold any more in the rectified case as discussed below.

##### Correlations Structure and Decorrelation

Equation (Equation 36) combines B cells RF overlap (in the matrix $C_{F}\left(s,s^{\prime }\right)$) to network effects, A cells and/or gap junctions, via the transfer operator $L^{\left(0\right)}$. One can actually better see these combined effects by projecting on the eigenvectors basis of $L^{\left(0\right)}$, where $L^{\left(0\right)}=P^{\left(0\right)}.\Lambda^{\left(0\right)}.{P^{\left(0\right)}}^{-1}$ and ${\tilde{L}}^{\left(0\right)}={\tilde{P^{\left(0\right)}}}^{-1}.\Lambda^{\left(0\right)}.\tilde{P^{\left(0\right)}}$. Denoting:(37)$$\Delta_{F}\left(s,s^{\prime }\right)={P^{\left(0\right)}}^{-1}.C_{F}\left(s,s^{\prime }\right).{\tilde{P^{\left(0\right)}}}^{-1},$$

Equation (Equation 36) becomes;
$$C_{\vec{X}}\left(t,t^{\prime }\right)=\int_{s=-\infty }^{t}\;\int_{s^{\prime }=-\infty }^{t^{\prime }}\;P^{\left(0\right)}.\;e^{\Lambda^{\left(0\right)}\left(t-s\right)}\;.\Delta_{F}\left(s,s^{\prime }\right).\;e^{\Lambda^{\left(0\right)}\left(t^{\prime }-s^{\prime }\right)}.\;\tilde{P^{\left(0\right)}}\;ds\;ds^{\prime },$$
which interprets as follows, whereas $C_{F}\left(s,s^{\prime }\right)$ is a rank $N_{B}$ matrix containing the B cell drives’ correlations, $\Delta_{F}\left(s,s^{\prime }\right)$ is a full rank matrix which integrates B cell drives and network correlations (due to the product with transfer matrices ${P^{\left(0\right)}}^{-1}$ and ${\tilde{P^{\left(0\right)}}}^{-1}$). These correlations are transported in time by the diagonal matrix $e^{\Lambda^{\left(0\right)}\left(t-s\right)}$. In general, there is no way to anticipate a priori what will be the combined effect of B cells RF overlaps and network on voltage correlations. Depending on the model parameters (characteristic times, synaptic weights), it can be anything. In particular, there is no general, mathematical reason, to think that A cells would decorrelate RG cell outputs.

This mathematical consequence is in apparent contrast with the claim, found in deep experimental papers stating that “the inhibition” (mediated by A cells) “decorrelates visual feature representations in the inner retina [15]”. What could be the origin of this discrepancy? The first reason is that correlations in the retina are often thought in terms of the drive correlations (Equation 33). Reducing the overlap between B cell RFs, i.e., decreasing the magnitude of the product $K_{B_{i}}\left(x-x_{i},y-y_{i},t-s\right)\;K_{B_{i^{\prime }}}\left(x-x_{i^{\prime }},y-y_{i^{\prime }},t^{\prime }-s\right)$ in the integral (Equation 33) lowers the drive correlations. The idea is then that A cells lateral inhibition reduces the center part of the RF and increases the surround, thereby reducing the RF overlap. Is there a way to mathematically validate this statement in (Equation 36)? Under which conditions on model parameters does it hold true?

Let us investigate what “decorrelation” means in our setting. Strictly speaking, it means that $C_{\vec{X}}\left(t,t^{\prime }\right)$ is diagonal that is that the variable change corresponding to the transfer matrix $P^{\left(0\right)}$ diagonalizes the stimulus correlation matrix $C_{F}\left(s,s^{\prime }\right)$. Now, $C_{F}\left(s,s^{\prime }\right)$, as a correlation matrix, is diagonalisable by an orthogonal basis change with real eigenvalues, whereas $P^{\left(0\right)}$ has to do with B cells–A cells network, and it is easy to find situation where it is complex, with complex eigenvalues. Thus, in general, the network effects do not diagonalize $C_{\vec{X}}\left(t,t^{\prime }\right)$. Nevertheless, it is indeed possible to construct networks diagonalising $C_{F}\left(s,s^{\prime }\right)$ by using the spectral decomposition theorem. In addition, if one does not stick to strict decorrelation, one can also figure out conditions on networks reducing stimuli correlations. The question is whether *real* A cell networks match these conditions. This is an interesting question for further studies. We, however, see below that there are, however, other potential sources of decorrelation, especially nonlinearities.

##### Non-Correlated Drives

The correlation structure, complex in the non-rectified case, is actually even worse when considering rectification. In the rest of this section, we want to consider in more detail the effects of rectification on RG cell spike correlations. We want to show that they induce non-stationary stimulus dependent correlations which *are not* due to the drives’ correlations (Equation 33).

For this, we are going to consider the situation where $C_{F}$ is δ-correlated that is we discard drive correlations. This corresponds to setting:(38)$$\vec{F}\left(t\right)=\vec{m}\left(t\right)+\sigma_{S}\xi \left(t\right),$$
where $\vec{m}\left(t\right)$ is deterministic. In this situation, Equation (Equation 36) greatly simplifies, giving a correlation matrix:(39)$$C_{\vec{X}}\left(t,t^{\prime }\right)=\sigma_{S}^{2}\;e^{L^{\left(0\right)}\left(t^{\prime }-t\right)}.\int_{-\infty }^{t}e^{{\tilde{L}}^{\left(0\right)}\left(t-s\right)}.e^{L^{\left(0\right)}\left(t-s\right)}ds,$$
for $t^{\prime }\geq t$.

In the general case, $L^{\left(0\right)}$ is not symmetric and does not commute with ${\tilde{L}}^{\left(0\right)}$. One can then compute $C_{\vec{X}}\left(t,t^{\prime }\right)$ in terms of the (common) spectrum of $L^{\left(0\right)},{\tilde{L}}^{\left(0\right)}$ using the spectral decomposition theorem $L^{\left(0\right)}=\sum_{\alpha =1}^{N}\lambda_{\alpha }^{\left(0\right)}\;v_{\alpha }^{\left(0\right)}.{\tilde{w}}_{\alpha }^{\left(0\right)}$ where $v_{\alpha }^{\left(0\right)}$ is the right eigenvector α of $L^{\left(0\right)}$ (the α-th column of $P_{}^{\left(0\right)}$) and ${\tilde{w}}_{\alpha }^{\left(0\right)}$ is the left eigenvector α of $L^{\left(0\right)}$ (the α-th row of ${\left(\;P_{}^{\left(0\right)}\;\right)}^{-1}$). In general, right (left) eigenvectors are not mutually orthogonal but ${\tilde{w}}_{\alpha }^{\left(0\right)}.v_{\beta }^{\left(0\right)}=\delta_{\alpha \beta }$ so that $v_{\alpha }^{\left(0\right)}.{\tilde{w}}_{\alpha }^{\left(0\right)}$ is the projector on eigendirection α. From this, one obtains the correlation matrix:(40)$$C_{\vec{X}}\left(t,t^{\prime }\right)=-\sigma_{S}^{2}\;\sum_{\alpha =1}^{N}\;e^{\lambda_{\alpha }^{\left(0\right)}\left(t^{\prime }-t\right)}\;v_{\alpha }^{\left(0\right)}.{\tilde{w}}_{\alpha }^{\left(0\right)}\;\sum_{\beta =1}^{N}\;\frac{v_{\beta }^{\left(0\right)}.{\tilde{w}}_{\beta }^{\left(0\right)}}{\lambda_{\alpha }^{\left(0\right)}+\lambda_{\beta }^{\left(0\right)}},$$
where eigenvalues are real or complex conjugate and are assumed to be stable (negative real part). Note that eigenvalues and projectors combine so that, finally, the correlation matrix is real.

We will keep this general form for further discussions on the rectified case, but here, it is insightful to consider the case where $L^{\left(0\right)}$ is symmetric. Here, it is diagonalizable on an orthogonal basis, with ${\left(\;P_{}^{\left(0\right)}\;\right)}^{-1}={\tilde{P}}^{\left(0\right)}$ and with real eigenvalues $\lambda_{\beta }\equiv -s_{\beta },\beta =1\cdots N$. where $s_{\beta }$ is real, positive. Then, (Equation 40) reduces, in form of components, to:(41)$$C_{\alpha_{2},\alpha_{1}}\left(t^{\prime }-t\right)=\frac{\sigma_{S}^{2}}{2}\sum_{\beta =1}^{N}\frac{P_{\alpha_{2}\beta }\;P_{\alpha_{1}\beta }}{s_{\beta }}\;e^{-s_{\beta }\left(t^{\prime }-t\right)}.$$

It is useful to express, from (Equation 41), the variance of cell $\alpha_{i_{1}}$’s voltage (independent of time due to stationarity):(42)$$\sigma_{\alpha_{1}}^{2}=\frac{\sigma_{S}^{2}}{2}\sum_{\beta =1}^{N}\frac{P_{\alpha_{2}\beta }\;P_{\alpha_{1}\beta }}{s_{\beta }}.$$

These computations provide the network correlations between cell voltage in the absence of drive correlations.

#### 3.2.2. Spike Correlations

We now compute spike correlations of RG cells induced by network correlations (Equation 40). We assume a spiking probability of the form (Equation 8). The probability that RG cell $\alpha_{1}\left(>N_{B}+N_{A}\right)$ spikes at time $t_{1}$ is induced by the voltage probability P and is given by $\nu_{\alpha_{1}}\left(\;t_{1}\;\right)\equiv E\left[\;f\left(\;\frac{V_{G}\left(t\right)-\theta_{G}}{\sigma_{G}}\;\right)\;\right]$, where the expectation is taken with respect to P. Taking the form (Equation 8) for *f*, this is:(43)$$\nu_{\alpha_{1}}\left(\;t_{1}\;\right)=f\left(\;\frac{m_{\alpha_{1}}\left(t_{1}\right)-\theta_{G}}{\sqrt{\sigma_{G}^{2}+\sigma_{\alpha_{1}}^{2}}}\;\right),$$
where $m_{\alpha_{1}}$ is the entry $\alpha_{1}$ of the deterministic drive term in (Equation 38). As pointed out above, two sources of noise add up here: the implicit noise, with variance $\sigma_{G}^{2}$ appearing in the LNP formulation (Equation 8), which is intrinsic to the cell, and the network induced noise, explicit in the term $\sigma_{\alpha_{1}}^{2}$.

Likewise, the probability that RG cell $\alpha_{1}\left(>N_{B}+N_{A}\right)$ spikes at time $t_{1}$ and RG cell $\alpha_{2}\left(>N_{B}+N_{A}\right)$ spikes at time $t_{2}$ is:(44)$$\nu_{\alpha_{1}\alpha_{2}}\left(t_{1},t_{2}\right)=\int \;f\left(\;\frac{\sqrt{\mu_{1}}\;\cos\left(\phi \right)\;y_{1}-\sqrt{\mu_{2}}\;\sin\left(\phi \right)\;y_{2}+m_{\alpha_{1}}\left(t_{1}\right)-\theta_{G}}{\sigma_{G}}\;\right)f\left(\;\frac{\sqrt{\mu_{1}}\;\sin\left(\phi \right)\;y_{1}+\sqrt{\mu_{2}}\;\cos\left(\phi \right)\;y_{2}+m_{\alpha_{2}}\left(t_{2}\right)-\theta_{G}}{\sigma_{G}}\;\right)\;DY,$$
where the integral holds on $R^{2}$ and where $DY=\frac{1}{2\pi }\;e^{-\frac{y_{1}^{2}+y_{2}^{2}}{2}}\;dy_{1}\;dy_{2}$. Here, $\mu_{1},\mu_{2}$ are the eigenvalues of the pairwise correlation matrix $C=\left(\;\begin{array}{cc} \sigma_{\alpha_{1}}^{2} & C_{\alpha_{i1}\alpha_{i2}}\left(t_{1}-t_{2}\right)\\ C_{\alpha_{i2}\alpha_{i1}}\left(t_{2}-t_{1}\right) & \sigma_{\alpha_{2}}^{2} \end{array}\;\right)$ which is diagonalizable on an orthogonal basis with an orthogonal transformation, a rotation with angle ϕ determined by the coefficients of C.

#### 3.2.3. Decorrelation Induced by Nonlinearities

It is evident that the double integral (Equation 44) factorizes only in the case where C is diagonal ($\phi =0,\mu_{1}=\sigma_{\alpha_{1}},\mu_{2}=\sigma_{\alpha_{2}}$), and it reduces to $\nu_{\alpha_{1}\alpha_{2}}\left(t_{1},t_{2}\right)=\nu_{\alpha_{1}}\left(t_{1}\right)\nu_{\alpha_{2}}\left(t_{2}\right)$. Thus, spikes of RG cell $\alpha_{1}$ at time $t_{1}$ and of RG cell $\alpha_{2}$ at time $t_{2}$ are decorrelated if and only if the correlation matrix (Equation 41) is diagonal. This matrix is diagonal only when there are no A cells. Otherwise, A cells have the effect of correlating voltages and thereby spikes. We already discussed above the possible effect of A cells in decorrelating the B cell drive term. Here, as we have removed this effect, we are in a position to discuss other potential effects inducing RG cell spike decorrelation.

First, note that, if the correlations we compute are non-vanishing, they can nevertheless be weak. The weakness of pairwise correlations in the retina has actually be reported by many authors [22,80]. It has been known since Ref. [92] that the passage of two correlated Gaussian variables through a subsequent nonlinearity always reduces the correlation of the two signals, regardless of the shape of the nonlinearity. Thus, in our case, the nonlinear function of the LNP model reduces the decorrelation.

Now, the LNP nonlinearity is not the only source of decorellation. Rectification also plays a crucial role. What happens, indeed, in the rectified case? Mathematically, one can use Equation (Equation 21) to compute the correlation matrices (Equation 40) (or even (Equation 36)), but the main, quite intricate problem is now that the entrance and exit time of domains $t_{-}^{\left(n_{k}\right)}$, $t_{+}^{\left(n_{k}\right)}$ appearing in (Equation 21) are *themselves* random. This is again a consequence of the stimulus dependence of these times. The computation of the voltage correlations in this case being, for the moment, out of reach, I am going to give some straightforward although insightful remarks.

The non-rectified case corresponds to a trajectory staying in the domain $\Omega^{\left(0\right)}$ (forgetting about conditions on noise ensuring that this holds for an infinite time). Now, the computation of voltage correlation is essentially the same if the trajectory stays in the domain $\Omega^{\left(n\right)}$. The only difference is that eigenvalues and projectors have a superscript (n) instead of (0). This difference is essential though because rectification induces a projection on the space of non-rectified cells. The contribution to rectified cells to voltage correlations with other cells vanishes thereby transforming the voltage correlation matrix. By permutation of rows and columns, one can convert this matrix in a form containing a diagonal block (correlations rectified cells ↔ rectified cells) and a block characterizing the correlations’ non-rectified cells ↔ as all cells. This reduces the model dimensionality and the global correlations. This effect, composed with the LN nonlinearity, can reduce correlations even more.

The last important remark here is that rectification implies that RG cell correlations are *stimulus dependent even if we have removed the drive correlations* because the exit times of continuity domains are stimulus dependent. In addition, the obtained correlations are non-stationary. This effect might not be noticeable with full field stimuli or white noise, which weakly solicit the lateral A cell connectivity, but it could be more prominent when studying spatio-temporal stimuli, in particular moving trajectories or non-stationary stimuli, which constitute most of the real visual scenes.

#### 3.2.4. Conclusions of This Section

In this section, we have mathematically investigated the structure of correlations induced by the model (Equation 13), Figure 1. Our conclusion is essentially that the stimulus generates RG cell spike correlations modulated, on one hand, by the drive correlations, and, on the other hand, by the B cells–A cells networks—more precisely, by the eigenvalues-eigenmodes of the transport operator. In addition, rectification and nonlinearities further impact correlations. This fact was reported by Pitkow and Meister in their paper “Decorrelation and efficient coding by retinal ganglion cells” [12], where they insist on the prominent role of nonlinearities: “Most of the decorrelation was accomplished not by the receptive fields, but by nonlinear processing in the retina”. From these remarks, they conclude about information transmission by the retinal network: “At very high thresholds, the information transmission is poor. Notably, transmission also drops at low thresholds. Thus, the choice of threshold involves a trade-off between rarely using reliable symbols, such as high spike counts, or frequently using unreliable symbols, such as low spike counts”. Thus, nonlinearities play a role in retinal coding making the spike rate of RG cells as sparse as possible, so that these cells are silent most of the time and fire at a high rate only when salient features of the stimulus make it necessary. This effect should be even more prominent for moving objects, which is clearly an example of a stimulus with salient features and strong spatio-temporal correlations induced by its trajectory, especially if this trajectory shows sharp changes. This could be mathematically analysed in the present setting although at the expense of consequent technical efforts.

Let us also remark that rectification makes the stochastic process of voltages non Gaussian because the times of entering and exiting domains are now random variables too. As a consequence, spike statistics involve higher order correlations. Although it has to be further investigated on experimental grounds, this would lead to important consequences in terms of coding. As pointed out, again, by Pitkow and Meister [12], “for highly non-Gaussian signals, such as neural spike trains and natural images, correlation may be only weakly related to redundancy.”

Sticking to the model, we may ask the following questions. Assume that we submit the model to different types of stimuli: the “classical” ones such as white noise, “Chirp” stimulus, natural images; but also more elaborated ones such as moving objects with different types of trajectories, or “natural movies” including motion and “surprise”—for example, a bird crossing the visual scene, with, on the background, a forest of trees in the wind. It is known that the retina is able to filter the “noisy” motion of tree leaves while signalling the bird, thanks to dedicated circuits involving A cells [1,56]. Such circuits can be easily implemented in the model (Equation 13) [45]. What will the structure of its spike trains be, depending on the different type of stimuli? How can one “efficiently” decode the stimulus from the mere knowledge of those spike trains? How efficient is a decoding scheme based on independent, decorrelated RG cells? In contrast, would cooperative network effects make the code more precise, affording faster responses to motion [14]?

Although we are not going to answer these questions here (there is still a long way to it), we give, in the next sections, several insightful mathematical results in this direction.

### 3.3. Computing the Mixed Effect of Network and Stimulus on Spike Correlations

#### 3.3.1. Context

Let us now consider the retina from the point of view of its output. We sit on the optic nerve and measure the spikes sent to the LGN and cortex via the optic nerve. We have no access to the biophysical machinery taking place in the retina and generating those spikes, but we know that the spike trains contain information about the external world stimuli that we want to extract. We can measure as many quantities as we want such as firing rate or higher correlations. More generally, we are seeking the (time dependent) joint probability of spikes adopting the approach described in Section 2.2, Methods.

In this context, assume a retina “at rest” i.e., receiving no stimulus or stationary stimuli like noise. We can describe the spike trains emitted by this retina by a stationary transition probability P, associated with a stationary probability $\mu^{\left(sp\right)}$ (for “spontaneous”). In general, this probability has spike correlations of order 2 and higher. Assume now that, from time $t_{0}$, a stimulus (say a moving object) is getting through the visual field of this retina. As exposed in Section 3.2, one expects the spike correlations (at any order) to be modified by this stimulation. Typically, a moving object carries spatio-temporal correlations in its trajectory which will superimposed upon the network correlations, resulting in a mixed effect where nonlinearities can also play a role. Can we predict, for a given stimulus, how correlations will be modified?

Let us give an example. Consider a linear chain of neurons, as depicted in Figure 3 (top). Each neuron (black points), is connected to its neighbours with an excitatory connection (red arrows) and to its second nearest neighbours with an inhibitory connection (blue arrows). The model here is a classical leaky integrate and fire model in the presence of noise, where parameters have been tuned to have a spontaneous asynchronous activity as depicted in Figure 3 (bottom,left). See [51] for more details. Consider a moving stimulus S(x,t) propagating from left to right (cyan, bell shaped curve) Figure 3 (top). S(x,t) acts as an input current of the form S(x,t)=f(x−vt), where *v* is the propagation speed and *f*, typically, a Gaussian. This stimulus is going to modify the spike patterns, as seen in Figure 3 (bottom,left), where one clearly sees nearest neighbours excitation and second nearest neighbours inhibition. The remarkable fact is that the stimulus not only modifies the firing rates of neurons, but also *their correlations*. The question is: can we compute this effect?

This question has been solved in the paper [51] for the gIF model (Equation 27). Here, we briefly state the main results (see the paper for technical details). Consider a function f(t,ω) (observable) depending on time and spike history up to time *t*. Let $\mu^{\left(sp\right)}$ be the join probability distribution of spikes in spontaneous activity (no stimulus), and μ the join probability distribution of spikes in the presence of a spatio-temporal stimulus S(x,t). We note $\delta \mu \left[\;f\;\right]\left(t\right)=\mu \left[\;f\;\right]\left(t\right)-\mu^{\left(sp\right)}\left[\;f\;\right]$, where $\mu \left[\;f\;\right]\left(t\right)$ is the average of *f*, at time *t*, in the presence of the stimulus and $\mu^{\left(sp\right)}\left[\;f\;\right]$ the average of *f* in spontaneous activity (which does not depend on time because spontaneous dynamics are stationary). $\delta \mu \left[\;f(t)\;\right]$ characterizes how much the time dependent mean of f(t,ω) under stimulation departs from the spontaneous mean at time *t*. In the simplest case, $\delta \mu \left[\;f(t)\;\right]$ characterizes the variation in the firing rate of neuron *k*, if $f\left(t,\omega \right)=\omega_{k}\left(t\right)$, or the variation in the correlation between neuron $k_{1}$ at time $t_{1}$ and neuron $k_{2}$ at time $t_{1}+t$ if $f\left(t,\omega \right)=\left(\;\omega_{k_{1}}\left(t_{1}\right)-\mu^{\left(sp\right)}\left[\;\omega_{k_{1}}\;\right]\;\right)\left(\;\omega_{k_{2}}\left(t_{1}+t\right)-\mu^{\left(sp\right)}\left[\;\omega_{k_{2}}\;\right]\;\right)$, and so on.

One can show that, when the stimulus amplitude is weak enough, $\delta \mu \left[\;f(t)\;\right]$ is given by a linear response formula of the form:(45)$$\delta \mu \left[\;f(t)\;\right]\;=\left[\;K_{f}*S\;\right]\left(\;t\;\right)$$

That is, by the convolution of the stimulus with a specific kernel, $K_{f}$, *depending on the observable f and on the spontaneous distribution $\mu^{\left(sp\right)}$*. We do not give the expression of this kernel here, for simplicity, but the reader can refer to the paper [51].

#### 3.3.2. Consequences

##### Convolution

Similarly to (Equation 1) (RG cells response to stimuli) or (Equation 2) (B cells response to stimuli), we have here again a linear response where the effect of a stimulus on a system is expressed by a convolution. We are, however, in a completely different perspective. Indeed, while we were formerly considering voltage response of individuals cells (shaped by network effects), we are now working on a more abstract level, where we attempt to measure the effect of a stimulus on *statistics*. This is of course due to the difference in what is accessible by experiments, what the observer is able to deal with in his observations—here spikes. Thus, the mathematical machinery allowing for extracting the response requires defining spike statistics in a non-stationary setting, where the influence of the stimulus can be inferred.

##### Kernel

The kernel $K_{f}$ can be explicitly computed in the gIF model. It depends on several features. First, on network characteristics (especially the effective interaction $W_{kj}$, and, more generally, the parameters shaping the model dynamics). It also depends on the observable *f*. However, the main content of this result is that the kernel $K_{f}$ is actually determined by spike correlations in *spontaneous activity*. In other words, it is possible to anticipate the response to a non-stationary stimulus from the knowledge of the spontaneous activity. Although this result is expected from Kubo theory in non-equilibrium statistical physics [93,94] or from Volterra–Wiener expansions [21], it has interesting consequences when dealing with neural dynamics, and more specifically here, with retina outputs. First, it provides a consistent treatment of the expected perturbation of higher-order correlations, beyond the known linear perturbation of firing rates and instantaneous pairwise correlations; in particular, it extends to time-dependent correlations. In addition, it reveals how the stimulus–response and dynamics are entangled in a complex manner. For example, the response of a neuron k to a stimulus applied on neuron i does not only depend on the synaptic weight $W_{ki}$ but, in general, on all synaptic weights because the dynamics create complex causality loops which build up the response of neuron k [49,95,96]. The linear response formula is written in terms of the parameters of a spiking neuronal network model and the spike history of the network. In the presence of stimuli, the whole architecture of synaptic connectivity, history and the dynamical properties of the networks are playing a role in the spatio-temporal correlations structure.

##### Linear Response and Higher Order Corrections

The derived formula provides a good agreement with simulations in the gIF model under time dependent stimuli (typically, a moving object). It requires, however, that the stimulus amplitude is weak enough. That is, higher corrections are weaker than the leading order. For larger amplitude stimuli, one would compute higher order correlations. This can be done using the same formalism [97], although it might not be the best approach. Indeed, this method requires measuring spontaneous correlations which are difficult to obtain experimentally for orders higher than 2. This is actually one of the reasons why LNP-like models exist. The expected nonlinearity in the response is handled by a static nonlinear function. Exploring what could be the best nonlinear correction to the linear response in such models is definitely an interesting mathematical challenge.

### 3.4. Conclusions

#### 3.4.1. Beyond Naive RF Description


This linear response theory actually shows how the neuronal network substrate and stimulus response are entangled. Indeed, in contrast to naive RF representation where the convolution kernel is assumed to depend only on the *cell|*, here, mathematics show that it depends as well on the *observable*. The explicit form of the kernel is also tightly constrained by the neurons’ connections. Finally, a convolution implies an integration over histories, requiring thereby to consider spike probabilities with memory, instead of “instantaneous” spikes probabilities (not or weakly depending on the past). Of course, one may always argue that, on experimental grounds, long tail memory is just impossible to measure so “instantaneous” [22] or first order Markov [98] models are largely sufficient. However, what does “sufficient” mean? This is a difficult question, which requires sophisticated methods to determine the “best performing” memory depth from data [34,99,100]. Actually, numerical computations of the kernel use, of course, Markovian approximations [51], although with a memory depth that can be controlled.

#### 3.4.2. Link with the Retina Model

Can we relate the formalism developed here with the retinal model presented in Section 2.1? As RG cell voltage is Gaussian, it is in principle possible to compute transition probabilities using the transport operator formalism. However, even in the non-rectified case, the computation promises to be a formidable task, unless one adds some additional constraints. For example, a big advantage of Integrate and Fire models is that a spiking neuron loses memory after spiking, a property which is not implemented in LNP like models.

#### 3.4.3. Information Geometry

There is a close link between Gibbs distributions and information geometry. This theory, developed by Shun’ichi Amari and his collaborators (see [101] and references therein) on the basis of early work from Rao [102], establishes a geometric theory of information where probabilities are considered as points on Riemannian manifolds. A prominent family of probability measures is called the exponential family. It contains the Gibbs distributions in the standard statistical physics sense, i.e., probabilities having the form $\frac{e^{-\beta H}}{Z}$ where the energy *H* does not depend on time. In this case, the metric is given by the Hessian of log(Z), the free energy, and is tightly linked to Fisher information on one hand and to linear response on the other hand. The linear response is actually a correlation function from the fluctuation dissipation theorem. Thus, correlation functions induce a natural geometry for Gibbs distributions providing strong insights on how these distributions are modified by smooth, local, transformations of their parameters (like learning [103]) or under a stimulation of weak amplitude. In this last case, the stimulus action corresponds to a perturbation in the tangent space of the manifold [104,105]. Although information geometry has not been extended, to our best knowledge, to the type of Gibbs distribution we study here (they are non-stationary), the mathematical formalism is similar. This essentially tells us that the structure of spatio-temporal correlations observed in spike trains reveals a hidden geometrical structure which, somewhat, shapes the response of the retina, and, henceforth of cortex, to stimuli. We come back to this point in the Conclusions section.

## 4. Applications

The OPL-B cells-A cells processing is based on graded potentials departing from the classical paradigm of binary spike processing. Mathematically, this has strong consequences in terms of response to a spatio-temporal stimulus: existence of eigenmodes, potentially modulated by nonlinear effects, inducing properties such as activity waves ahead of the stimulus (anticipation), resonances and correlations modified by the stimulus. In this section, and although this paper is essentially theoretical, I would like to shortly propose possible applications of these results, outside the field of neuroscience.

### 4.1. Retinal Prostheses

Retinal pathologies, such as Age Macular Degeneration or Retinitis Pigmentosa, are due to the degeneration of photo-receptors [106]. In addition, they induce morphological and structural changes in the retina with significant pathogenic effects: inflammation, change in connectivity, the appearance of large-scale spontaneous electrical oscillations, and, of course, attenuation of response to visual stimuli [107,108,109,110]. In this process of degeneration, however, the RG cells are the last to be deficient, maintaining, therefore, a link between the retina and the brain, provided they are suitably stimulated. The strategy of retinal prostheses is to stimulate the retina electrically by an array of electrodes. Stimulation of an electrode generates, in the visual cortex, a phosphene, the perception of a light spot. By stimulating the electrodes, one induces in the cortex an image “pixelised” by the phosphenes, with resolution limited, on the one hand, by the number of electrodes, and, on the other hand, by the size of the phosphenes, which can be enlarged by diffusion and nonlinear effects [111]. Technological solutions, taking into account the physiological limitation on the electrical power that can be injected in an electrode, improve resolution [112]. However, there are still obstacles which cannot be resolved by purely technological solutions (hardware). In addition, a valid stimulation strategy at a given period of the pathology may not be later because the retina degeneration evolves over time.

Stimulation strategies use processor pre-processing to calculate, from a given image (captured by a camera), the pattern of stimulation of the prosthesis, by mimicking the calculation that a healthy retina would make, or by incorporating corrections taking into account the pathology [113]. These algorithms might be improved using what we know about the retinal structure, especially A cells’ lateral connectivity, where a model like (Equation 13) can be easily implemented with a relatively low energy consumption cost. The idea would be to improve electrode stimulation sequences in order to allow an implanted patient to perceive in real time a moving object. The model (Equation 13) with A cells’ lateral connectivity and gain control is known to produce a wave of activity ahead of a stimulus, performing a form of anticipation [45]. This could be used to compensate for the processing times imposed by the equipment, in the same way that the visual system knows how to compensate for the delays induced by photo-transduction [42]. The ideal would also be to have adaptive algorithms, i.e., depending on parameters adjustable according to the patient and the course of his pathology.

### 4.2. Convolutional Networks

Several recent studies attempt to understand how retinal response to stimuli is related to circuit processes using convolutional neural network models [114] to grasp the structure of retinal prediction [115]. Reciprocally, these networks can be used to design deep-learning models to encode dynamic visual scenes with important potential outcomes in the domain of computer vision. In particular, a recent work by Zheng et al. [116] shows the important role played by recurrence in encoding complex natural scenes. To my best knowledge (which is quite scarce in this field), there is no mathematical analysis of the dynamics of these models, especially the dependence on parameters and robustness of the training schemes. The present study could bring some insights on this perspective. Even if the model (Equation 13) is different from what these researchers were using, the techniques of piecewise linear phase decomposition and eigenmode study could be insightful to better understand the dynamical evolution of these convolutional networks and the role played by rectification.

## 5. Discussion

In this paper, we have addressed mathematically the potential effect of A cells lateral connectivity on retinal response to spatio-temporal stimuli. We have seen how, mathematically, the retina structure and the collective dynamics of retinal cells organized in local circuits spanning the whole retina might constrain this response. In particular, the structure of correlations is expected to depend on the stimulus, as soon as nonlinear effects are involved. This goes beyond the expected effect of stimulus correlations induced by RF overlap.

These properties are established on the basis of theoretical results which are based on incomplete modelling of the retina and specific assumptions. Their validation would require experiments, some of which may require time and others are not yet accessible, for example, simultaneously measuring retina and cortex. As a matter of fact, one may argue that the models presented here are far too simplistic compared to the real retina(s) having a large number of B cells, A cells and RG cell types, making complex circuits [18] and whose characteristics depend, in addition, on species, age or pathologies. However, the idea behind mathematical modelling is precisely to try and infer some generic mechanism underlying the real object under study, here the retina. This is the simplicity of the structure which makes it generic. The question is: “Would the addition of more elaborated retinal features make the response to stimuli simpler?”

In the next section, I discuss some further implications of this work leading to some new questions.

### 5.1. Cortical Response

If a dynamical stimulus, combined with the retinal network and nonlinearities, produces non-negligible dynamical spatio-temporal correlations, what could be the consequences at the cortical level? (For simplicity, I am going to consider the LGN as a simple relay). There is a physiological transformation, called retinotopy, which maps smoothly the retina topology to the cortical V1 topology. In models, it is usually considered to be the identity map, although it is not. This is a nonlinear transformation, depending, in addition, on the species [117,118,119]. Nevertheless, what matters here is that this mapping is smooth and invertible. Therefore, retinotopy transports, in a smooth and invertible way, the spatio-temporal retinal correlations to the visual cortex. This leads to a question: “How can a cortical model taking into account spatio-temporal spike correlations be defined?”

Cortical models are usually based on mean-field approximations where one features firing rates evolution, but not spike correlations. This is the case of the Wilson–Cowan model [120,121,122] or neural field models [123,124,125]. I know about two mean-field approaches taking care of spike correlations.

The first approach is the one initiated by S. El Boustani et A. Destexhe [126] using a Markovian approach to write down mean field equations of second order (i.e., including pairwise spatial correlations) and a non static thalamic entry that can feature the retinal-LGN input. This model can be used to construct a retino-cortical model [127], although the mathematical consequences of having correlated retinal entries have not been explored yet.

The second approach is based on the so-called Ott-Antonsen Ansatz [128] and has been used by Montbrio, Pazo and Roxin to propose an exact mean field approach with second order statistics [129]. Since their paper, there has been a lot of activity in developing this model, especially in connection with cortical imaging, with impressive results [130,131,132,133]. It is a promising track.

All these approaches could certainly provide powerful numerical and mathematical tools to better understand how spatio-temporal retinal correlations could be processed. In particular, having a retino-(LGN)-cortical model allows for doing a task that is currently impossible experimentally: measuring simultaneously the retina and cortex.

### 5.2. Retinal Correlations and Neurogeometry

We have also seen that retinal correlations and Gibbs distributions naturally define a metric on a Riemannian manifold where probabilities are points on this manifold. In particular, the application of a weak amplitude stimulus corresponds to a perturbation along the tangent space of this manifold. What is the image of this metric under the retinotopic transformation? Let us make this question a bit more precise.

The visual system has evolved to map as efficiently as possible retinal output to cortical structures. The shaping of the visual system during development is actually a highly dynamical process involving retinal waves and synaptic plasticity [88]. These processes provide the visual system a structure allowing it to respond in a fast and efficient way to the stimuli coming from the external world, via the retina. In particular, the capacity of the visual cortex to respond to spike trains with spatio-temporal correlations induced by natural stimuli should be somewhat imprinted in the cortical connectivity.

Visual perception is actually highly geometrically structured and shaped by the structure of cortical connectivity. This leads researchers to introduce a link between the geometry of cortex and the geometry of vision in the concept of neurogeometry (or neuromathematics) where the functional architecture of V1 is considered as a Lie group of symmetry with a Riemannian geometry (see [52,53,54,55] and reference therein). In this approach, cortical columns are point-like processors detecting visual features where functional connectivity is represented in terms of geodesics. To my best knowledge, neurogeometry essentially deals with V1 and static percepts, although extensions to motor cortex [134] and motion areas [135] have been done. Now, a natural question is: “Is there a relation between the cortical metric of neurogeometry and the metric induced by spatio-temporal spike correlations observed by the retina?”.

Let us address the problem the other way round: Projecting the cortical metric back to the retina via the inverse retinotopy map, what do we find? Is there a physiological correspondence with the retina structure and especially lateral connectivity? What could be the consequences of spike train statistics and on the way retina processing visual stimuli?

What do cortical metrics tell us about retinal spike correlations? Dealing with neural coding of vision, the simplest assumption consists of assuming that RG cells are independent encoders and that the cortex makes the job of restoring the spatio-temporal correlations exist in the visual scene (e.g., in the trajectory of a moving object). The alternative proposition, where spatio-temporal correlations imprinted in the RGC spike trains are deciphered by the cortex, makes the question of stimuli decoding by the cortex more challenging, but opens up far more possibilities. Answering to these questions could be based, as a first step, on important results existing in the literature—in particular, recent works asking the extent to which retinal connectivity and dynamics affect higher order features later derived from its outputs (e.g., orientation, spatial frequency speed etc) in V1 via LGN [136,137,138].

## Figures and Tables

**Figure 1 jimaging-08-00014-f001:**
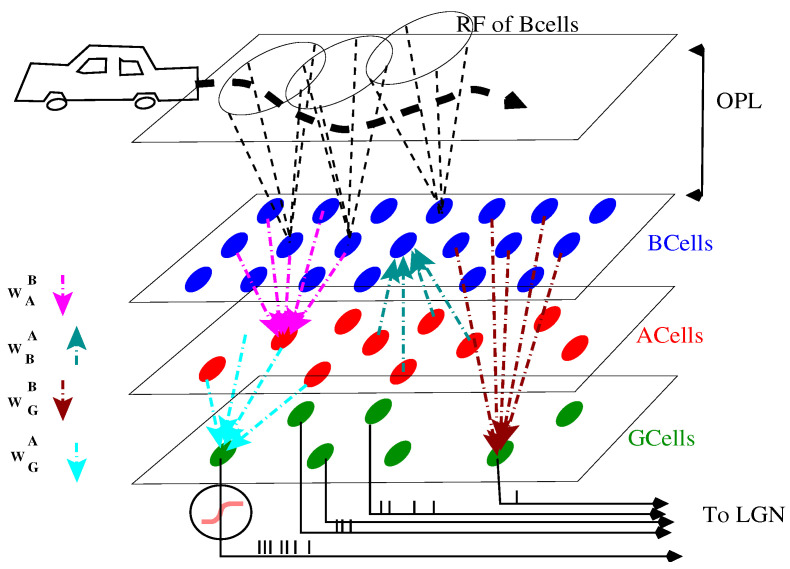
Structure of the retina model introduced in Section 2.1. A moving object (here, presumably, a car) moves along a trajectory (dashed black line). Its image is projected by the eye optics to the upper retina layers (Photoreceptors and H cells) and stimulates them. In the model, this corresponds to the convolution of the stimulus with the Receptive Field (RF) of B cells. This provides to B cells what we call the “OPL” input to B cells. B cells (blue points) are connected to A cells (red points) via excitatory synapses (pink arrows, denoted $W_{A_{}}^{B_{}}$) and to RG cells (green points) via excitatory synapses (brown arrows, denoted $W_{G_{}}^{B_{}}$). A cells are connected to B cells via inhibitory synapses (green arrows, denoted $W_{B_{}}^{A_{}}$) and to RG cells via inhibitory synapses (cyan arrows, denoted $W_{G_{}}^{A_{}}$). The voltage of RG cells is sent through a nonlinearity (pink curve in the black circle) so as to produce spike trains conveyed to the LGN.

**Figure 2 jimaging-08-00014-f002:**
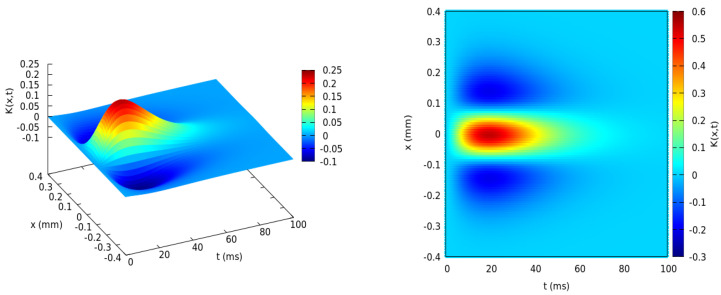
Receptive Field of a ON BCell. (**Left**) Example of a spatio-temporal RF of B cells (ON center cell) represented in 3D (one dimension of space, *x* and time *t*). There is inhibition at the surround, physiologically due to H cells. (**Right**) Spatio-temporal RF representation with a color map.

**Figure 3 jimaging-08-00014-f003:**
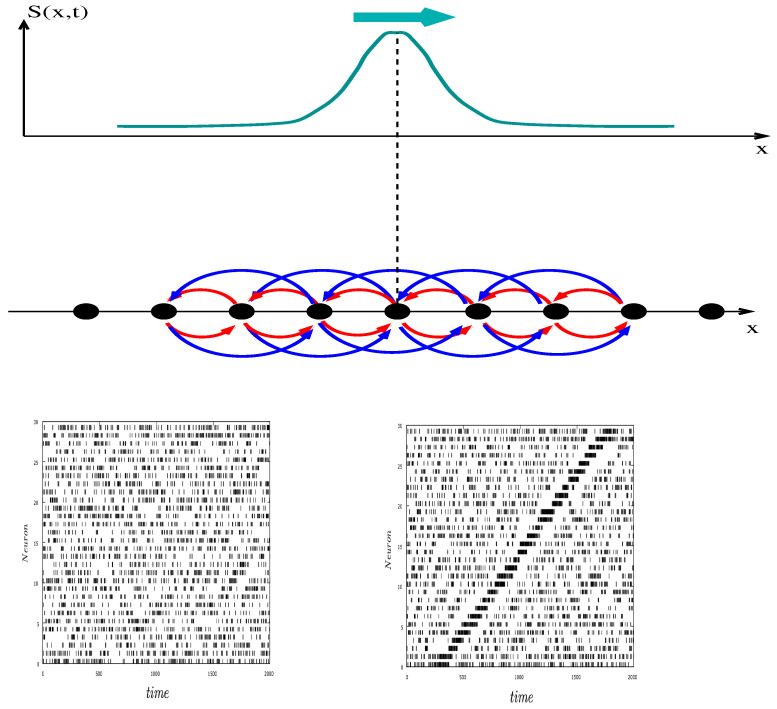
(**Top**) Network of spiking neurons sensing a stimulus (Redrawn from ref. [51]). Each neuron, represented as a black point, is connected to its neighbours with an excitatory connection (red arrows) and to its second nearest neighbours with an inhibitory connection (blue arrows). In addition, each neuron is able to sense external stimuli S(x,t) (cyan, bell shaped curve). (**Bottom Left**) Spontaneous spiking activity. (**Bottom Right**) Spiking activity in the presence of the moving stimulus.

## Data Availability

Not applicable.

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
