# Peer review of "Retinal Processing: Insights from Mathematical Modelling"

_2313-433X, 2022, doi:10.3390/jimaging8010014_

Round 1

Reviewer 1 Report

This is a dense modeling/mathematical paper aimed at deriving retinal ganglion cell correlation structure in the context of amacrine cell network architecture and input. This builds off the author’s previous work and explores the weak input, linear response regime of the model.

Major points:

1) A claim is made that there is very little modeling of amacrine cell dynamics. While this is certainly a poorly understood component of the retinal machinery, some references to the work that has been done on amacrine cells would be useful here. (e.g. Taylor, Demb, Singer…)

2) It seems a bit off to talk about gain control in the introduction, then punt on that in the methods and point either to other papers or future work, given that gain control has been invoked several times to effectively model the complex and surprising processing that the retinal carries out in response to moving stimuli (reversal response, motion anticipation, etc.). This should be clarified in the introduction.

3) No mention is made of some important large-scale retinal simulations that precede this work (notably the work of Wohrer). This work should be cited and discussed. It would also be worth contextualizing this work in terms of common GLM models of the retina and more biophysically detailed models, e.g. from the Berens group.

4) In Section 2.2.1 on voltage correlations, the choice of stimulus dynamics that only reflect noise should be discussed a bit more. It seems that the conclusion that the cell-cell correlation function is stimulus independent only holds under very restricted conditions, for example when the stimulus itself is not correlated in time and space except through this noisy OU process. This seems at odds with some of the grand exposition in the introduction.

5) Section 2.2.2 launches into a commentary on and long quotations from a paper from Meister and Pitkow and should be condensed to make the main point of that work in the author’s own words.

6) Results of simulations are pointed to in the results, but not shown. Those should be shown.

7) Parameter choices for these simulations should be listed somewhere in the paper.

8) The link to information geometry is tenuous and could be omitted from the paper.

Minor points:

There are some grammatical and usage errors in the paper, many to do with the agreement (singular or plural) between the noun and the verb in a sentence. These should be carefully corrected in proof before publication.

Retina ganglion cells are usually abbreviated “RGCs” in the literature. The decision to use GCell as the abbreviation, as well as HCell, BCell, etc, seems a bit unnecessarily jarring for the experienced reader. Using RG cell instead might be slightly more familiar.

It would be good if the abstract contained somewhat more detail about the conclusions drawn by the work.

The introduction is engaging but lacks sufficient references and development of some ideas. Most notably, there are a few “throwaway” statements in the second paragraph of the intro, lines 22-25, alluding to how the retina’s power efficiency might pertain to global warming. If it remains, this should be built out a bit more and supported by more concrete connections. There are certainly major concerns (and articles written) about the energy consumption of modern machine learning algorithms, like transformers. Referencing these more specific examples (or others) would ground the connections hinted at in this paragraph.

Figure 1, the left image is grainy and is largely redundant with the right diagram. It seems unnecessary to link to a textbook picture of the retina, but if that is okay with this journal, that’s fine.

Line 64-65 [typo] “neuro[-]biologists experts”, one or the other should be omitted

Line 104: It seems wrong to assert that amacrine cells, via common input to ganglion cells, produce an effective “causal” interaction. Common input creates correlation, but it’s then explicitly not causal. Surely this is a typo.

Line 119: “…all what the LGN and cortex see” should be “…all that the LGN and cortex see”

Line 194: “retina’s like structure” should be “retina-like structure” or “retinal structure”

Line 229: “because [they are] hardly accessible…”

Line 274: A bit more detail about the input, F(t), would be very useful. One thing that was unclear was if any assumptions are made about the correlation structure of the input, either in space or time. Without that info, it seems that the model can only be understood from a mathematical perspective (i.e. analytically) in a single time step, for a single stationary input value. Since F is clearly nonstationary, this should be made a bit more clear.

Line 283: There are well-documented gap junctions between RGCs and this should be stated and cited here (see e.g. Awatramani as well as many others in the mouse retina). It should be clearly stated that these exist but are not modeled in this study.

Throughout the methods, the equation formatting and exposition is very clear and easy to follow.

Before equation 14: it would be nice to have a sentence explaining C(t) just a bit more explicitly, in words.

Line 370: A little more about how the stimulus is modeled here would be illuminating. e.g. if the stimulus is strong, how can the statement that the \chi(t) trajectory is dominated by the network characteristics be valid?

~line 371: Some intuitive explanation for H^k_m would be nice.

Section 1.2 largely reviews previous work by the author, and could be condensed/summarized.

Line 678: “physicists” should be “physicist’s”

Line 852: this line seems unnecessary

Reviewer 2 Report

See pdf
